



# Developing a deep learning forecasting system for short-term and high-resolution prediction of sea ice concentration

Are Frode Kvanum[1,2], Cyril Palerme[1], Malte Müller[1,2], Jean Rabault[3], and Nick Hughes[4]

[1]Development Center for Weather Forecasting, Norwegian Meteorological Institute, Oslo, Norway
[2]Department of Geosciences, University of Oslo, Oslo, Norway
[3]IT Department, Norwegian Meteorological Institute, Oslo, Norway
[4]Ice Service, Norwegian Meteorological Institute, Tromsø, Norway

**Correspondence:** Are Frode Kvanum (arefk@met.no)

**Abstract.** There has been a steady increase of marine activity throughout the Arctic Ocean during the last decades, and maritime end users are requesting skillful high-resolution sea ice forecasts to ensure operational safety. Different studies have demonstrated the effectiveness of utilizing computationally lightweight deep learning models to predict sea ice properties in the Arctic. In this study, we utilize operational atmospheric forecasts as well as ice charts and sea ice concentration passive microwave observations as predictors to train a deep learning model with ice charts as the ground truth. The developed deep learning forecasting system can predict regional sea ice concentration at one kilometer resolution for 1 to 3-day lead time. We validate the deep learning system performance by evaluating the position of forecasted sea ice concentration contours at different concentration thresholds. It is shown that the deep learning forecasting system achieves a lower error for several sea ice concentration contours when compared against baseline-forecasts (persistence-forecasts and a linear trend), as well as two state-of-the-art dynamical sea ice forecasting systems (neXtSIM and Barents-2.5) for all considered lead times and seasons.

## 1 Introduction

Arctic sea ice thickness and extent have decreased since the first satellite observations were obtained (Kwok, 2018; Serreze and Meier, 2019) as a response to climate change (Notz and Marotzke, 2012) which is amplified in the Arctic region (Serreze and Barry, 2011). Summer months are experiencing the greatest loss of sea ice extent (Comiso et al., 2017), with models from the Coupled Model Intercomparisson Project 6 (CMIP6) projecting the first virtually ice-free (< 1 million square km) Arctic summer before 2050 (Notz and Community, 2020). As a consequence of the sea ice retreat during the summer months, there has been an increase in maritime activity in the Arctic (Eguíluz et al., 2016; Gunnarsson, 2021) resulting in a consistent increase in the number of ships present in the Arctic. The period during which many vessels operate has also extended beyond the summer months, increasing mariners exposure to hazardous sea ice conditions (Müller et al., 2023). The influx of operators to the Arctic region has increased the demand for accurate short-range sea ice forecasts (Stocker et al., 2020), and that end-users needs are taken into account during the validation of these forecasts (Melsom et al., 2019; Wagner et al., 2020).

Although dynamical sea-ice forecasting systems have been producing operational forecasts at different resolutions and lead times e.g. (Sakov et al., 2012; Metzger et al., 2014; Williams et al., 2021; Röhrs et al., 2023), feedback from maritime operators



suggests that current sea ice forecasts lack sufficient and relevant verification (Veland et al., 2021). Consequently, maritime
operators tend to rather rely on their own experience (Blair et al., 2022) despite the improved situational awareness provided by
sea ice forecasts for tactical navigation (Rainville et al., 2020). Moreover, dynamical forecasts are computationally expensive,
especially when targeting high spatial resolutions. In recent years, statistical forecasting approaches have emerged where deep
neural networks have been trained on past sea ice information as well as the state of the atmosphere in order to predict the future
state of sea ice concentration (SIC) (e.g. Fritzner et al., 2020; Liu et al., 2021b; Andersson et al., 2021; Liu et al., 2021a; Ren
et al., 2022; Grigoryev et al., 2022). These machine learning approaches require little memory and computational resources to
produce a forecast, once they are trained.

     Previous studies (Liu et al., 2021b; Andersson et al., 2021; Liu et al., 2021a; Ren et al., 2022) train deep learning models
on reanalysis datasets such as ERA5 (0.25° resolution) (Hersbach et al., 2020) or use SIC derived from coarse resolution (25
km resolution) satellite climate data records (such as the products from Cavalieri et al. (1996) and Lavergne et al. (2019)).
Andersson et al. (2021) proposed IceNet, a pan-Arctic U-Net which consistently improved upon the seasonal numerical fore-
casting system SEAS5 (Johnson et al., 2019) for lead times of 2 months and longer. Similarly, Liu et al. (2021b) showed that
a convolutional LSTM covering the Barents Sea with a 6 week lead time was more skillful than persistence for all considered
weekly lead times. However, due to the aforementioned models using climatological-scale data as predictors and ground truth,
their application to maritime users as short term operational forecasts are limited (Wagner et al., 2020).
Grigoryev et al. (2022) presented a multi-regional U-Net forecasting system for lead times up to 10 days where the real-
time availability of SIC satellite retrievals and numerical weather forecasts were considered. The deep learning forecasts of
Grigoryev et al. (2022) considerably outperformed persistence and linear trend baseline forecasts in the considered regions of
the Barents, Labrador, and Laptev Seas. Fritzner et al. (2020) demonstrated the possibility of utilizing a deep learning system
to forecast sea ice for the region around Svalbard and the Barents Sea, however the forecasts had a coarse spatial resolution
due to limited computational resources. High resolution sea ice forecasts are important for this region as it is the focus of many
commercial operators from different maritime sectors such as shipping, fishing and tourism (Stocker et al., 2020; Müller et al.,
2023).

     In this paper we present the development of a regional deep learning forecasting system targeting 1 km spatial resolution
and 1 – 3 day lead time covering the area around Svalbard and the Barents Sea. The choice of predictors and target data is
made with operational concerns, and the quality of the forecasts is assessed against relevant baseline forecasts and dynamical
sea ice forecasting systems in a manner relevant for end-users (Melsom et al., 2019; Wagner et al., 2020). The impact from the
different predictors is also assessed. Section 2 describes the datasets used for this study, followed by Section 3 presenting the
neural network implementation and verification setup. Section 4 presents the results, with Section 5 providing the discussions
and conclusions.





**Table 1.** Products used, their application as well as temporal regime. Observational products and physical forecasting models are separated by descriptive italic text. Time regime refers to which time period the dataset covers with respect to the initialization date of the deep learning model.

| Product | Variables | Training | Validation | Time regime |
|---|---|---|---|---|
| *Observations* | | | | |
| Ice charts | SIC | Predictor / Target | Yes | Present / Future |
| OSI SAF SSMIS | SIC trend | Predictor | Yes | Past |
| AMSR2 (ASI) | SIC | No | Yes | Future |
| *Models* | | | | |
| AROME Arctic | T2M, X,Y-Winds | Predictor | No | Future |
| NeXtSIM | SIC | No | Yes | Future |
| Barents-2.5 | SIC | No | Yes | Future |

## 2 Data

To develop the deep learning forecasting system, several observation and physical model forecasting system datasets have been chosen as predictors, targets and for validation. When selecting appropriate datasets, their spatial resolution as well as release frequency has been considered in order to develop an operational product. Table 1 presents the different products we have used, as well as the role they play in our forecasting system which will be further described in the following sections. The region of interest is depicted in Fig. 1 and is constructed as an intersection between the regional domains of the gridded ice chart data produced by the Norwegian Ice Service (https://cryo.met.no/en/latest-ice-charts) and the regional numerical weather prediction system AROME Arctic (Müller et al., 2017). The deep learning model has been developed using the U-Net architecture (Ronneberger et al., 2015), which requires the spatial dimensions of the input fields to be repeatedly divisible by a given factor a number of times. For simplicity, the model domain was set to be a one kilometer spatial resolution square grid containing $1792 \times 1792$ equidistant grid-cells, which is four times divisible by $4$. This domain was achieved by removing lower latitudes from the original AROME Arctic domain, affecting the Norwegian, Barents and Kara seas.

### 2.1 Sea-ice concentration observations

The ice charts are manually drawn to deliver a SIC product which is distributed every workday at 15:00 UTC by the Ice Service of the Norwegian Meteorological Institute (https://www.cryo.met.no/en/latest-ice-charts). The ice analyst who draws the ice chart assesses and merges available synthetic aperture radar (SAR) scenes with visible- and infrared imager observations. These data sources are supplemented by coarse resolution passive microwave observations to achieve a consistent spatial coverage. We use gridded SIC from the ice charts as both a predictor representing current sea ice conditions and a target at $1 - 3$ day lead time since the product captures daily (weekdays from Monday to Friday) observed SIC at a high spatial resolution. The



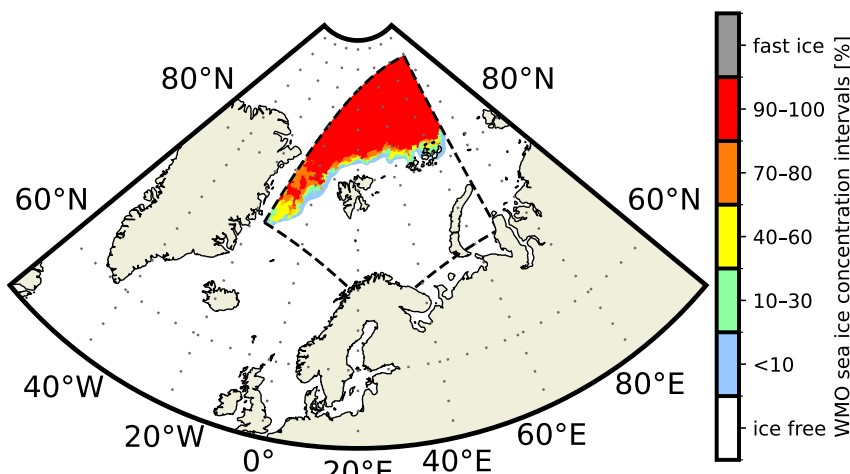

**Figure 1.** The model domain (dashed-contour) together with the SIC retrieved from a ice chart (15 sep 2022). The SIC intervals and color code follows the WMO Ice Chart Colour Standard and Sea Ice Nomenclature.

ice charts are a categorical product, with SIC following the World Meteorological Organization (WMO) total concentration
intervals (see colorbar of Fig. 1). For this study, the ice charts have been gridded onto the model domain with a 1km spatial resolution. Moreover, we have filtered out Baltic sea ice as the task of the deep learning system in this study is to predict sea ice in the Greenland and Barents seas.

In addition to the ice charts, SIC observations from the Ocean and Sea Ice Satellite Application Facility (OSI SAF) Special Sensor Microwave Imager/Sounder (SSMIS) (OSI-401-d) and AMSR2 observations processed with the ASI algorithm from
the University of Bremen (Spreen et al., 2008) are utilized. OSI SAF SSMIS is supplied on a 10km spatial resolution, and will be used to compute a linear sea-ice concentration trend which will serve as both a predictor and as a baseline-forecast for validation. Motivated by the lack of temporal awareness of the U-Net architecture (Ronneberger et al., 2015), computing a linear trend from past sea-ice concentration fields will encode multiple previous time-steps into a single two dimensional field. Moreover, computing the linear trend from a different product than the ice charts will supply the model with correlated but not
overlapping information. It is also noted that the ice charts are not produced every day, hence it would not be possible to use the product to compute a local trend.

The AMSR2 observations are used for validation of the deep learning forecasting system only. The AMSR2 data utilized for this work is the ASI sea-ice concentration product from the University of Bremen (Spreen et al., 2008). The dataset is provided on a 6.25 km grid. The AMSR2 observations are fully withheld from the data used to train the deep learning model. Hence,



the AMSR2 data are used as an external product for validation of forecast performance, providing an estimation of the deep learning model's ability to provide consistent forecasts beyond using the ice charts as validation.

## 2.2 Physical forecasting systems

In addition to training the deep learning model on current and previous sea-ice concentration data, we also include atmospheric predictors as it has been demonstrated that the inclusion of the present- and future state of the atmosphere can improve the sea-

ice predictions from deep learning (Grigoryev et al., 2022; Palerme et al., 2023). For this study, forecasts of 2-meter temperature and the 10-meter wind components adjusted to align with the x,y dimensions of the model grid (x,y-wind) were taken from the AROME Arctic regional numerical weather prediction system developed for operations at the Norwegian Meteorological Institute (Müller et al., 2017). Although not a forecast field, the land-sea mask used in AROME Arctic is also extracted as a predictor. We use AROME Arctic forecasts as predictors for this study due to its high spatial resolution while also covering

most of the ice chart domain. AROME Arctic runs up to 66 hours lead time, is supplied on a 2.5 km resolution grid with 66 vertical levels, and a new forecast is initiated every six hours. Near surface winds influence the sea ice drift following a non-linear relationship between wind speed, sea-ice drift speed, sea-ice concentration and sea-ice thickness (Yu et al., 2020). Moreover, near surface temperatures affects the sea-ice through melting or growth. AROME Arctic has been in operation and continuous development since October 2015. However due to a major change of the representation of snow over sea-ice in

2018, which significantly lowered a warm bias of near-surface temperatures above sea-ice in the model (Batrak and Müller, 2019), the training dataset consists of dates starting from 2019 in order to use a consistent dataset.

Moreover, the two short-range sea-ice forecasting systems neXtSIM-F (Williams et al., 2021) and Barents-2.5 (Röhrs et al., 2023) are used to validate the deep learning forecasts against high-resolution physical forecasting systems. neXtSIM-F is based on the neXtSIM sea-ice model which is a dynamical/thermodynamical sea-ice model using a brittle rheology (Rampal et al.,

2016). The version of neXtSIM used for this work uses the Brittle Bingham Maxwell rheology (Ólason et al., 2022), and is supplied on a pan-Arctic grid at 3 km resolution. Barents-2.5 is a regional ocean and sea-ice ensemble forecasting system developed at the Norwegian Meteorological Institute (Röhrs et al., 2023), and is produced on a 2.5 km spatial resolution and runs up to 66 hours lead time on the same grid as AROME Arctic. The sea-ice model used in Barents-2.5 is CICE (Hunke et al., 2015). At prediction time, six members are initiated, with one member receiving atmospheric forcing from AROME Arctic and

the rest by atmospheric forecasts from ECMWF, however for this study only the member forced by AROME Arctic has been considered. Finally, due to recent developments of the model, only forecasts starting from June 2022 have been considered from Barents-2.5.



## 3 Methodology

### 3.1 Dataset preprocessing and selection

We perform preliminary computations in order to ensure that the input data from different sources are on a common grid. The data preprocessing is performed in two stages. Firstly data not matching the AROME Arctic projection are reprojected. Secondly, for data available at a coarser resolution, nearest neighbor interpolation is performed in order to resample the data onto a 1 km grid. The U-Net architecture requires all predictors to have valid values in all grid cells, however both the ice charts and SIC trend do not consistently represent SIC for land covered grid cells due to their intended unavailability. In order

to avoid sharp gradients between sea-ice covered seas and land covered areas, we apply a nearest neighbor interpolation of the local sea-ice conditions to fill in the missing sea-ice concentration over land grid points following Wang et al. (2017).

  Since all the datasets we use come from operational products, we have to take into account production- and publishing time as well as forecast length when selecting predictors. A graphical summary of the operational schedule for predictor selection is shown at the top of Fig 2. The ice charts are published every workday at 15:00 UTC, which will be regarded as the initialization

time for the deep learning forecasts. The OSI SAF linear trend is computed from the five previous days, until the day before deep learning forecast initialization. Since we want the AROME Arctic forecasts to provide information regarding the future state of the atmosphere, it follows that the forecast should cover the time after an ice chart has been published. Moreover, the ice charts are drawn based on data available until their publication time. This provides a temporal limit for the atmospheric forecasts when an ice chart is used as target during training, since the lead time of the data gathered from the atmospheric

forecasts should not exceed the publication time of the target ice chart (15:00 UTC).

  We choose to use AROME Arctic forecasts initiated at 18:00 UTC on the same day as the ice chart publication. Furthermore, we set the reference time of AROME Arctic forecasts to be 12:00 UTC the day of prediction regardless of model lead time of 1, 2 or 3-day. This way, we ensure that atmospheric forecast cover the time period in between ice chart publication and intended target lead time. Moreover, AROME Arctic initiated at 18:00 UTC reaches 12:00 UTC for a 3-day target lead time after 66

hours, which motivates the choice of having 12:00 UTC as reference time regardless of target lead time. In addition, AROME Arctic has a timeliness of 2.5 hours, which ensures that the forecast initiated at 18:00 UTC allows us to publish deep learning forecasts on the same day as the input ice chart is published.

  Selecting atmospheric forecasts starting at 18:00 UTC causes atmospheric development between 15:00 and 18:00 to be missed by the network. Although AROME Arctic is also initiated at 12:00 UTC, the forecast initiated at 18:00 is more up to

date, and as such is assumed to be more reliable especially at longer lead times. Finally, the ice charts do not represent the sea ice state at the time they are published, rather they are a mean representation of previous observations accumulated over time ending at publication time. Hence we assume regardless of AROME Arctic initialization time that there will be some irreducible timing difference between the sea ice state from the ice charts and the initial atmospheric state from AROME Arctic, which also varies spatially.

Instead of loading multiple high-resolution AROME Arctic fields as predictors, we have explored predictor reformulations which reduce the amount of memory needed to load predictors. We reduce the atmospheric forecast fields between start-date





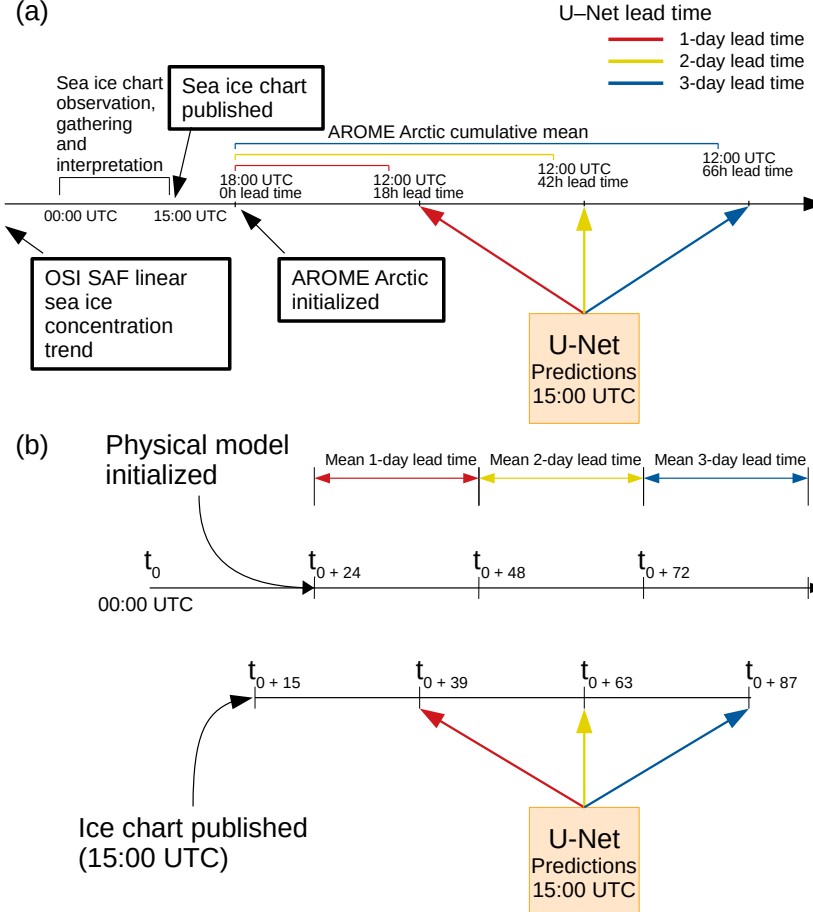

**Figure 2.** Overview diagrams describing a) predictor publication scheduling, selection and preprocessing, and b) model initialization for intercomparison. a) Description of when the different predictors are published in relation to a published ice chart when constructing a single sample for a given date. The ice charts are published at 15:00 UTC, followed by AROME Arctic initialized 18:00 UTC (available ~ 20:30 UTC). The different colors refer to deep learning forecast lead time. b) Description of the forecast intercomparison setup between physical forecasting systems with an hourly frequency and deep learning forecasts. The timestamps are given relative to 00:00 UTC the day an ice chart is published. The color code above is reused.

and 12:00 UTC at target date along the temporal dimension into a mean field. As well as reducing the memory footprint of each predictor, reducing the time steps into a mean-value field also accumulates the temporal changes of each atmospheric variable into a single predictor. Supplying the predictors as single fields compared to time-series is also appropriate when



**Table 2.** Subset affiliation and number of samples for each year over the different target lead times.

| year | subset | 1-day lead time | 2-day lead time | 3-day lead time |
|------|--------|-----------------|-----------------|-----------------|
| 2022 | test | 196 | 147 | 142 |
| 2021 | validation | 198 | 147 | 142 |
| 2020 | train | 198 | 146 | 142 |
| 2019 | train | 192 | 143 | 144 |

considering that the U-Net architecture is not designed to exploit structured sequences of data (Ronneberger et al., 2015). As a consequence, deep learning models targeting different lead times are trained independently, and the atmospheric predictors cover an increasing temporal range as a function of lead time.

The main dataset we use covers the period between 2019 and 2022. We further split the data such that 2018 – 2019 is used for training, 2020 is used for validation and 2021 is the test dataset. Table 2 provides an overview of the number of available
samples for each year given each model target lead time. Moreover, the predictors are normalized according to the min-max normalization equation. This normalization scheme ensures that the different predictors are in the same numerical range $[0,1]$ and that predictors can be drawn from non-normal distributions such as the ice charts. Finally, with this scheme we can combine categorical predictors from the ice charts with continuous predictors from AROME Arctic.

Due to the routinely lack of ice charts during weekends, there is a limited number of dates that can be used for training
and verification, and the sample size depends on lead time as shown in Table 2. Comparing the similarly sized 2 and 3-day lead time datasets against the number of samples at 1-day lead time reveals an approximate 25% reduction in the number of available dates consistent for all considered years. This has implications when the ice charts are used to evaluate deep learning forecast performance because verification scores for models targeting different lead times get computed from different sets of dates.

Motivated by the skewed SIC category distribution between the categories which constitutes the marginal ice zone (MIZ) $(10 – 30\%, 40 – 60\%, 70 – 80\%)$ and the "very close drift ice" $(90 – 100\%)$ category (as visible in Fig. 1), we reformulate the target SIC such that each category is defined cumulatively and predicted independently. Cumulative contours are a novel reformulation of the SIC prediction task. Since each sea ice category represents a range of SIC, each cumulative contour contains SIC equal to or greater than a given SIC contour (lowest boundary of the WMO SIC category). Hence each cumulative
contour represents a more balanced prediction task, than each individual category.

The cumulative contours are defined as follows. Let $C \in \mathbb{R}^3$ be a set representing $(N > 2)$ contour elements with spatial indexes $i, j$ and elements $c_{i,j}^n$. Moreover, let $S \in \mathbb{R}^2$ represent a sea-ice chart, with $x_{i,j}$ being the sea-ice concentration values between 0 and 1. Then, let $k_n \in [0,1]$ be thresholds

$$0 \leq k_1 < k_2 < \cdots < k_n \leq 1. \tag{1}$$





Hence, each cumulative contour is defined as

$$c_{i,j}^n = \begin{cases} 1 & \text{if } s_{i,j} \geq k_n \\ 0 & \text{if } s_{i,j} < k_n. \end{cases} \tag{2}$$

The target reformulation into cumulative contours reduce the classification task into multiple independent binary predictions. Each cumulative contour has a spatial extent comparable to the sea ice extent, hence the categories in the MIZ are not under-represented in the dataset. Finally, the forecast SIC field is defined as the sum over all cumulative contours:

$$\text{Forecasted sic} = \sum_{\text{for all } n} c^n. \tag{3}$$

## 3.2 Model implementation

The U-Net architecture was initially developed for computer vision tasks, specifically semantic image segmentation, and expands the fully convolutional architecture introduced in Long et al. (2015) by constructing a symmetric encoder-decoder structure and adding skip-connections between the contracting and expansive paths (Ronneberger et al., 2015). Our U-Net im-

plementation follows the original encoder-decoder structure, however the output layer has been modified in order to reflect the reformulated target SIC cumulative contours. The encoder is initiated with 64 feature maps, and we established through testing that the model performed optimally with a depth of 256 feature maps in the bottleneck. This results in a three stage encoder, where the spatial resolution is lowered by a factor of four at both steps due to the average pooling at the end of each stage. Note that the average pooling layer used here deviates from the max-pooling layer used in the original U-Net architecture, as

we found through tests that average pooling tended to increase model performance similar to the findings from Palerme et al. (2023). We further note that in the original U-Net architecture the spatial resolution of the feature maps are only lowered by a factor of 2 between each stage, however the current implementation reaches the bottleneck faster which further reduces the size of the models. As a consequence of reformulating the target variables as cumulative contours, the model contains as many output layers as there are cumulative contours, such that each cumulative contour is predicted independently from a shared

signal. The pixelwise binary cross-entropy loss function is computed individually for all layers, and the resulting loss of the model is the sum over the individually computed losses. We initiate the model weights using HE-initialization (He et al., 2015) since the ReLU activation function (Nair and Hinton, 2010) is used for all layers. All models have been trained on a NVIDIA A100 GPU using mixed precision training, which restricted the maximum batch size to four samples to fit in the GPU RAM. Consequently, we replace all batch-normalization layers in the encoder and decoder with group-normalization layers to miti-

gate the negative effects of using batch-normalization with small batch sizes (Wu and He, 2018). During training, we use the ADAM optimizer (Kingma and Ba, 2014) with an initial learning rate of 0.001 with a piecewise weight decay reducing the learning rate by a factor of 2 every 10 epochs. After training is completed (25 epochs), the model which achieves the lowest loss on the entire validation set is selected. The flow of data in relation to the developed model is summarized in Figure 3. For further details regarding the implementation, we refer to the GitHub repository (see code availability section).



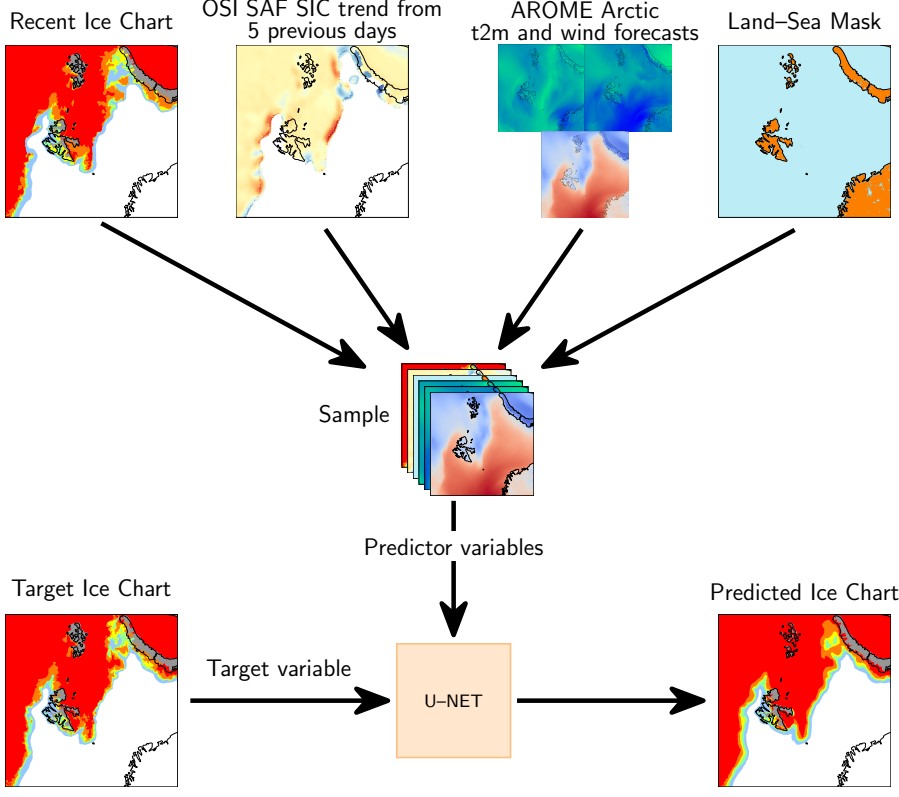

**Figure 3.** Overview of the input and output to the deep learning forecasting system. The predictors are constructed from individually preprocessed sources, and provided to the network together with an associated target ice chart.

## 3.3 Verification metrics

We chose to focus on sea ice edge based skill metrics when validating the performance of the deep learning forecasts as such metrics are appropriate when the SIC is discretized as categorical contours. These metrics are also relevant for end users (Melsom et al., 2019; Fritzner et al., 2020; Wagner et al., 2020). Specifically, we derive the length of the sea ice edge following the method introduced in Melsom et al. (2019), and assess forecast skill using the Integrated Ice Edge Error (IIEE) (Goessling et al., 2016) normalized with the ice edge (or threshold SIC contour) length derived from the target SIC field (nIIEE). The nIIEE is chosen since it is not particularly affected by isolated ice patches (Palerme et al., 2019). Moreover, the nIIEE when normalized according to a SIC contour length, is independent to the sea ice seasonality (Goessling et al., 2016; Palerme et al., 2019; Zampieri et al., 2019), which allows for a comparison of forecast-skill across seasons. Finally, the nIIEE can be interpreted as the SIC contour displacement error between two products, which is easy to interpret and relevant to end-users (Melsom et al., 2019). To the knowledge of the authors, the nIIEE has only been assessed using coarse resolution sea-ice concentration fields. However, we compared the nIIEE computed from ice charts at 1 km spatial resolution and 10 km





resolution between 2019 and 2022 and found the Pearson correlation to be 0.98, which ensures the validity of applying the nIIEE also for high-resolution SIC. For further details, see the appendix.

## 3.4 Baseline-forecasts

We compare the deep learning forecasts against two baseline-forecasts, persistence of the observations and a linear trend of sea-ice concentration from OSI SAF SSMIS. The baseline-forecasts serve as a lower threshold which the deep learning system must outperform in terms of nIIEE in order to be considered skillful. A persistence forecast involves keeping the initial state of the system constant in time. The baseline-forecast based on the linear trend is created by computing a pixelwise linear trend from the previous 5 days which is used to advance the system forward in time. For clarity, the computed values are bounded

to match the valid value range $[0, 100]$. The use of a linear SIC trend as a baseline forecast has previously been assessed in Grigoryev et al. (2022), where the authors reported that the linear trend consistently achieved a higher Mean Absolute Error than persistence.

## 3.5 Model intercomparison setup

The goal of the model intercomparison is to assess the predictive skill of the deep learning forecasts against the described

baseline-forecasts and physical forecasting system. In order to compare the different sea ice forecasts, all products were projected and interpolated onto the grid of the coarsest resolution product, which is neXtSIM (3 km) or AMSR2 (6.25 km) depending on which SIC product is used for evaluation. Both baseline-forecasts have a daily output frequency, which is similar to the deep learning system, hence the comparison involves identifying the forecast with similar start- and target date. However, both Barents-2.5 and neXtSIM forecasts have an hourly frequency. When comparing the deep learning forecasts against both

physical models, we use the physical forecasts initiated at 00:00 UTC the day following deep learning initialization. Furthermore, a daily mean is computed from the model timesteps which covers the target date of the deep learning forecast. This setup is assumed to moderate the spatial variability induced by the lack of a temporal mean. The intercomparison setup is presented lowermost in Fig. 2.

## 4 Results

### 245 4.1 Training performance and data considerations

Training the deep learning system for 25 epochs takes approximately 3h30min on the A100 GPU, whereas performing a single prediction takes 6 seconds on a workstation CPU (AMD EPYC 7282 16–Core) and 30 seconds on a laptop CPU (Intel (R) Core (TM) i7–8565U 8–Core). The optimal U–Net depth of 256 channels in the bottleneck was determined by performing a gridsearch across learning–rate and U-Net depth. The final model contains 2.4 million trainable parameters with 1.15 million

of these being located in the encoder and 1.25 million in the decoder. We compared model implementations without cumulative contours (single output, multi–class segmentation) against deep learning models reformulated with cumulative contours, and





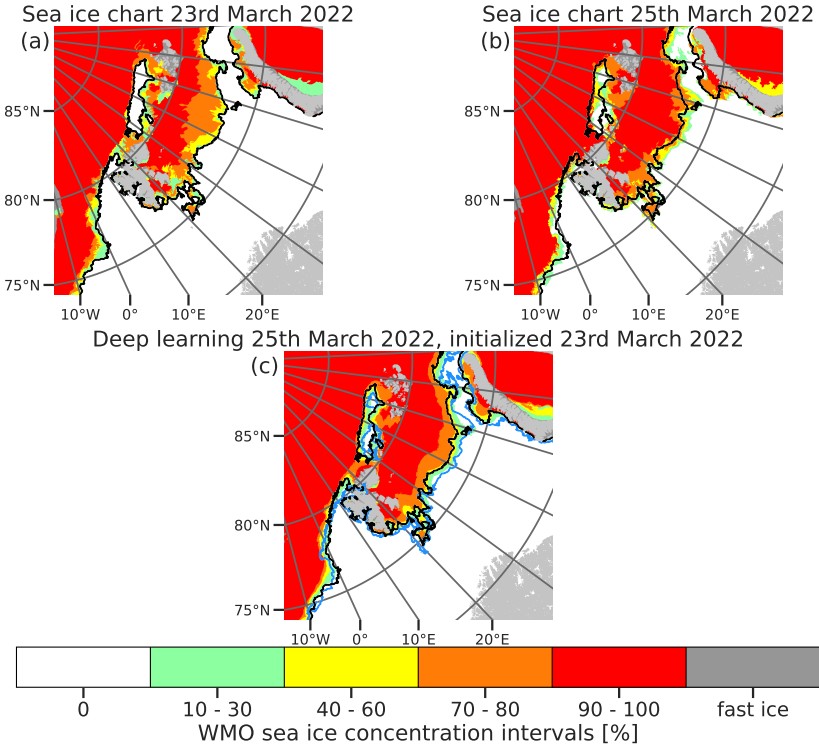

**Figure 4.** Ice charts for the 23rd (a) and 25th (b) of March 2022, with a deep learning prediction for 25th of March 2022 initialized 23rd of March 2022 in (c). The black line is the sea ice edge for the ice chart in (a) and the blue line is the sea ice edge for the ice chart in (b), both plotted for a 10% concentration threshold. The < 10% SIC category is not shown.

found that the prior did not predict the *very open drift ice* (10 – 30%) and *open drift ice* (40 – 60%) SIC categories for any samples in the test–dataset. This is probably due to the low number of pixels belonging to these categories compared to the other classes. Fig. 4 presents a forecast from a deep learning model with cumulative contours targeting 2–day lead time, and

shows that both *very open drift ice* and *open drift ice* have been resolved in the forecast. For the example presented in Fig. 4, the deep learning forecast achieved an nIIEE of 7.5 km while persistence achieved an nIIEE of 13.4 km. We observe in Fig. 4 that the deep learning forecast is able to reproduce the SIC increase in the Barents Sea, as well as the reduction of a polynya area north-east of Svalbard. An apparent difference between the deep learning forecast and the ice charts is that the different contours include less structural details in the deep learning forecasts, which results in a smoother appearance.

Fig. 5 compares the ability of the deep learning system to resolve sea ice categories against ice charts and AMSR2 observations. In general, the deep learning system accurately resolves the concentration category distribution in accordance with the ice charts regardless of lead time, with all categories being less than 1% different from the ice chart distribution when considering the yearly average. When comparing against the AMSR2 observations, it is important to note the differences in the occurrence frequency of the 100% SIC category. The ice charts consider fast-ice as a separate category representing land-fast





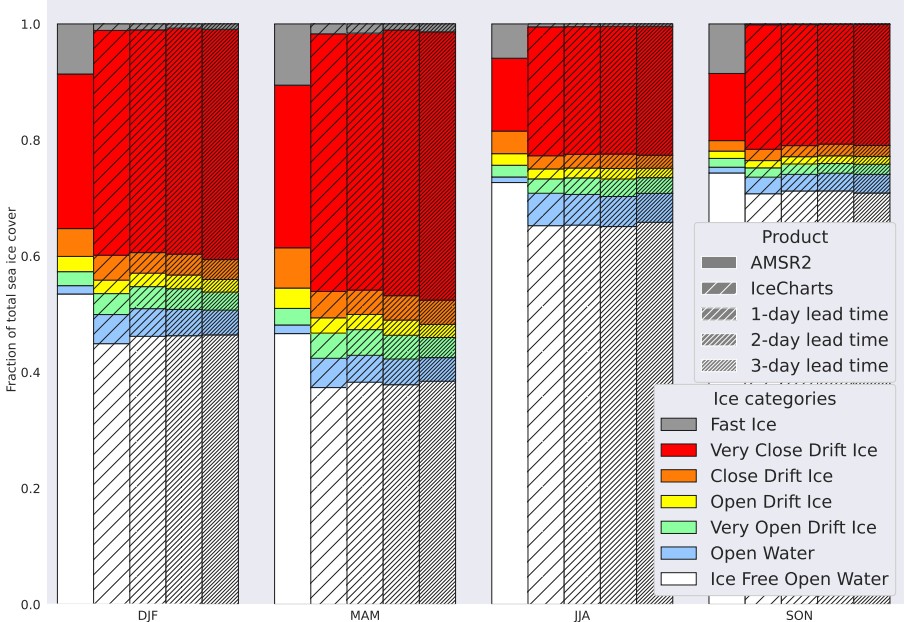

**Figure 5.** Seasonal distribution of each SIC category for 2022 as respective fraction of the total mean SIC area for AMSR2, ice charts and the deep learning system at $1 - 3$ day lead time. The AMSR2 data have been projected onto the deep learning model domain.

ice, which is a distinction not made by the ASI retrieval algorithm. Although for consistency, 100% SIC from AMSR2 has been considered as *Fast ice* for this study. However the normalized integrated ice edge error only consider the lower boundary of any concentration category and as such this choice does not affect the results from the nIIEE skill score. This choice is reflected in Fig. 5 where the resolved fraction of *Very close drift ice* is 20% in AMSR2 compared to 31% in the ice charts. Comparatively, the fraction of resolved *Fast ice* in AMSR2 is 8% whereas for the ice charts this category constitutes $< 1\%$ of the area.

Another difference between AMSR2 observations and the ice charts presented in Fig. 5 is how *Ice free open water* and *Open water* are resolved. On a yearly average, *Ice free open water* constitutes about 62% of the AMSR2 pixels, and 55% for the ice charts. Furthermore, *Open water* is more represented in the ice charts constituting about 5% of the pixels, while for the AMSR2 observations this category covers only 1%. This is because the ice charts consider SAR and optical satellite retrievals with higher sensitivity to low ice concentrations to resolve *Open water*, compared to passive microwave sensors which have a 275  low sensitivity to SIC below 15%.

## 4.2 Forecast performance and model intercomparison

We initially compare the deep learning forecasts against the baseline and dynamical forecasts in 2022 across all target lead times where we consider the yearly mean of the nIIEE for a sea ice edge defined with the 10% SIC contour in Fig 6. For all considered lead times, the deep learning forecasts achieves the lowest nIIEE. Similar to persistence, nIIEE for the deep learning 280  forecasts increases proportionally with lead time, although at a lower rate. Additionally, neither neXtSIM nor the linear trend



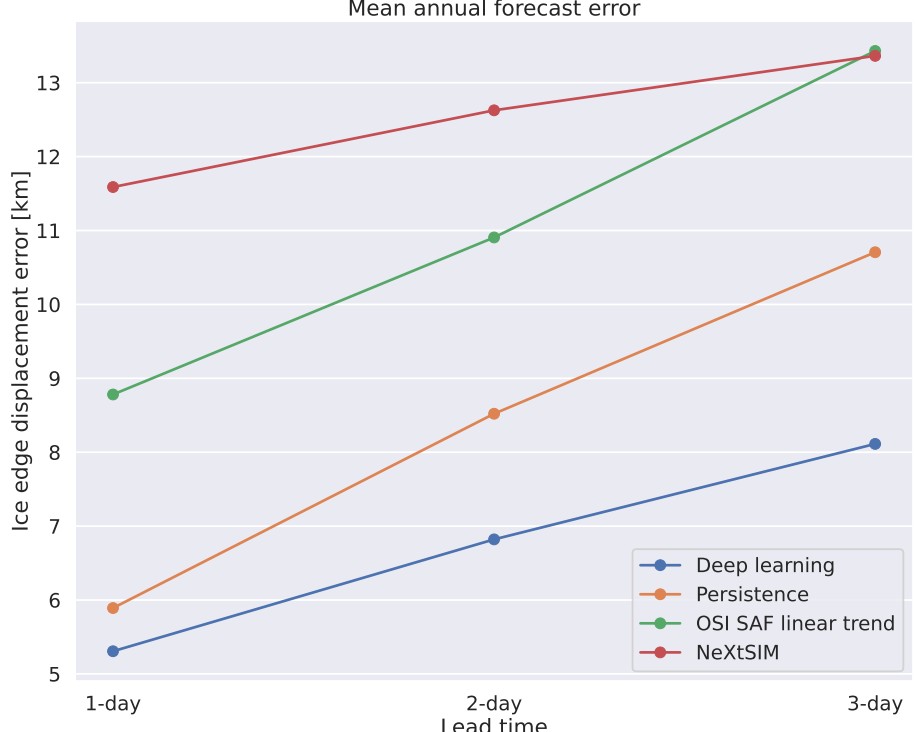

**Figure 6.** Mean annual ice edge displacement error as function of lead time. The sea ice edge has been defined by the 10% concentration contour. Only products with a complete coverage of 2022 have been considered.

forecast are able to outperform persistence, on average scoring a factor of 1.57 and 1.34 worse than persistence, respectively. Comparatively, the deep learning forecasts improve upon persistence by reducing the nIIEE by a factor of 0.82. In terms of error-growth as a function of lead-time, the linear trend forecast is the only forecast where the slope of the error increases with increasing lead-time. This indicates that the linear trend from past OSI SAF SSMIS observations is unable to capture ice

chart evolution especially for longer lead times. Moreover, the neXtSIM forecasts have the lowest error-growth with lead-time, indicating that neXtSIM may provide more useful forecasts at longer lead-times.

    Fig. 7 shows how the deep learning system resolves the seasonal variation of the sea ice edge length for different lead times. The predicted sea ice edge follows a similar seasonal pattern to the ice edge length from the target ice charts. Each monthly mean predicted sea ice edge length has a negative bias compared to the ice charts, which increases for longer lead times. Given

that the deep learning forecasts resolve the different categories akin to the ice charts, we attribute the apparent negative bias of the length to the lack of details along the forecast contour edges. Hence SIC contour smoothness is somewhat proportional to forecast lead time.

    In order to assess the consistency of the deep learning forecasts trained on ice charts, we evaluate the performance by replacing the ice charts with AMSR2 observations as reference dataset in Fig. 8. When utilizing AMSR2 observations as

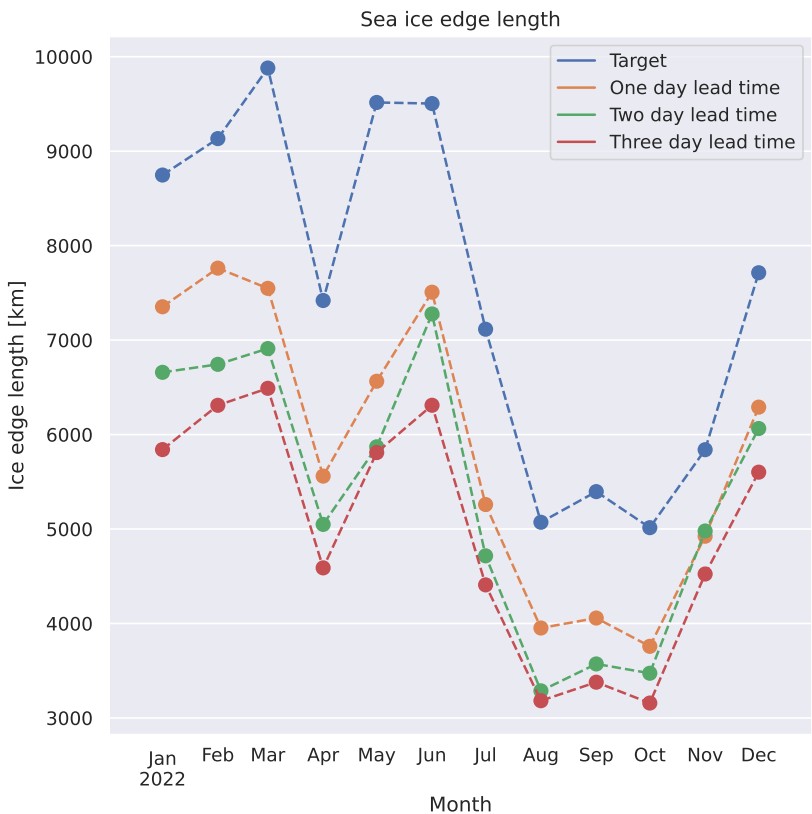

**Figure 7.** Mean monthly sea ice edge length for 2022, with the sea ice edge defined by a 10% concentration threshold. The considered products are the ice charts and deep learning system for 1 − 3 day lead times.

reference, the number of samples used to evaluate the forecasts is consistently 247 across all lead times. We see in Fig. 8 that the deep learning forecasts on average achieves the highest nIIEE when considering a 10% concentration contour, achieving a mean nIIEE of 16.5 km across the lead times. The displacement is consistent with the inherent nIIEE difference between the AMSR2 observations and the ice charts (Fig. 5), which we found to be 13.3 km for the 10% concentration contour when compared across the test dataset. Furthermore, AMSR2 persistence forecasts achieves the lowest nIIEE on average for the same contour. When considering SIC contours defined by $\geq 40\%$ SIC, the deep learning forecasts perform closer to AMSR2 persistence, albeit achieving a slightly higher nIIEE on average. neXtSIM on average outperforms the deep learning forecasts for the 10% concentration contour, however this is not the case for the 40%, 70% or 90% concentration contours. For the contours higher than 10% SIC, Fig. 8 shows that both AMSR2 persistence and the deep learning forecasts on average gradually improve against both neXtSIM and the linear trend, with the deep learning forecast increasing its improvement against neXtSIM for higher contours. Overall, AMSR2 persistence mostly achieve the lowest nIIEE, although surpassed by the deep learning



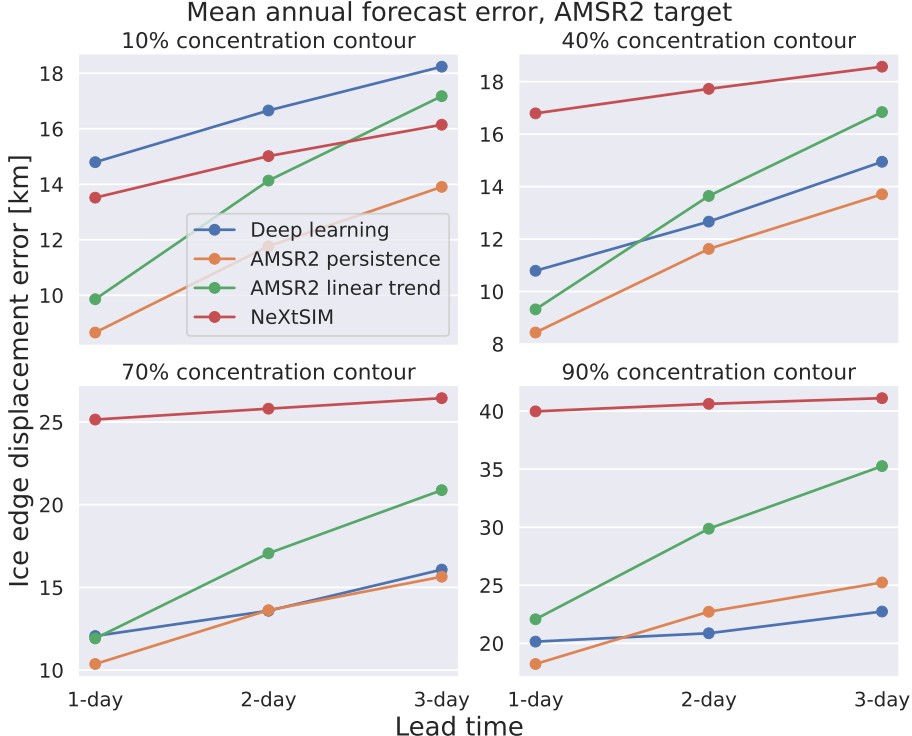

**Figure 8.** Mean annual ice edge displacement error as a function of lead time. The ice edge displacement error for the different products has been computed considering AMSR2 observations as reference.

forecasts when higher concentration contours $>= 90\%$ and $>= 2$-day lead time are considered. Moreover, the deep learning forecasts achieve the lowest nIIEE scores when targeting the 40% concentration contour from the AMSR2 observations, in good agreement with the average nIIEE difference between AMSR2 and the ice charts which we found to be 9.7 km for the same concentration contour.

The model intercomparison experiment which compares the deep learning system against baseline- and dynamical sea ice forecasts is presented in Fig. 9 using the ice charts as reference. For all considered lead-times and target contours, the deep learning forecasts achieve the lowest seasonal mean nIIEE. The seasonal axis of Fig. 9 shows that both ice chart persistence and the deep learning forecasts achieve higher nIIEE values during winter and spring, associating the errors to the periods of freeze-up and sea ice maximum extent. When the nIIEE is computed from the 70% or 90% concentration contours, Fig. 9 shows that

the forecasts not utilizing ice chart information (i.e. linear trend, neXtSIM and Barents-2.5) attain considerably higher values, especially during summer. This pattern might indicate a discrepancy between the ice charts, the dynamical forecasts and linear trend with regards to how higher SIC is resolved, further influenced by seasonal conditions.



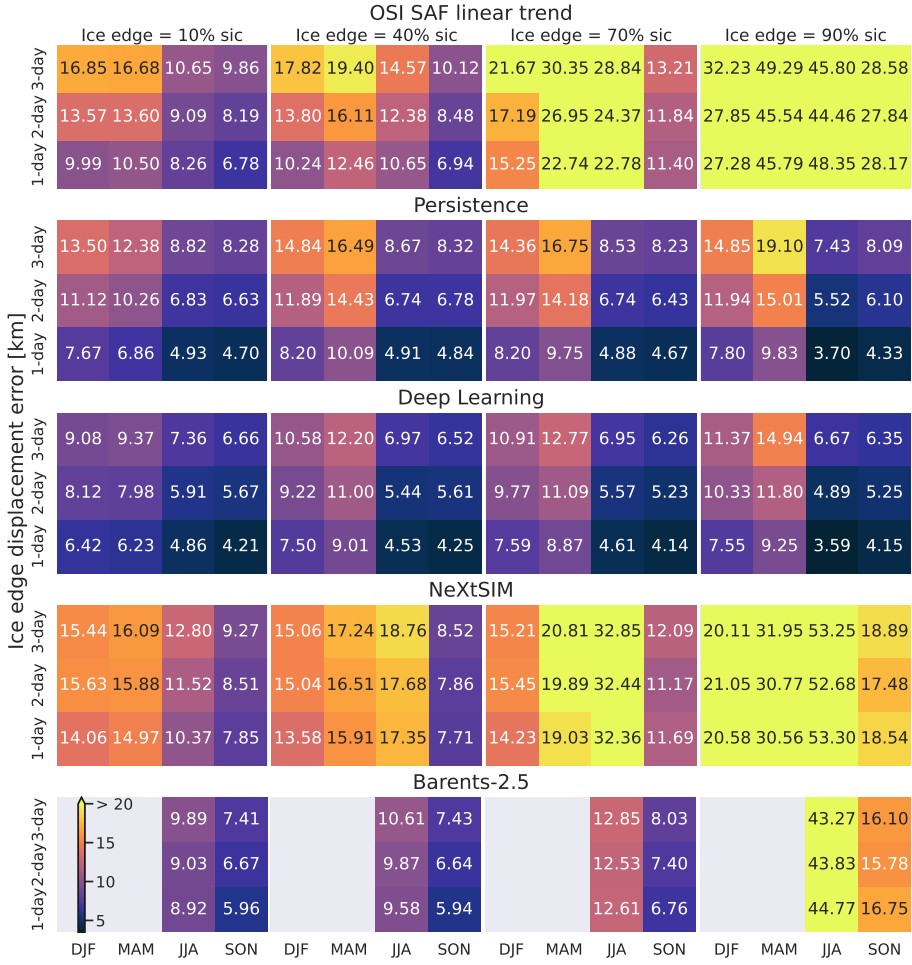

**Figure 9.** Model intercomparison for varying seasons, lead times and concentration contours. The ice charts are considered as reference. The values reported represent the integrated ice edge error normalized according to the length of the current SIC contour from the reference ice chart in km. The OSI SAF linear trend is computed from the past five days. Barents-2.5 results are only shown for summer and fall.

### 4.3 Feature importance

To better understand the importance of the different predictors used, as well as the sensitivity of the deep learning system to the predictors, we measured how the model responds to modified predictors. In order to measure the impact of each predictor, we first conducted an experiment where the nIIEE was computed from deep learning models fitted to different predictor subsets. The effect of including different predictors on deep learning forecast performance is shown in Fig. 10. In general, removing predictors tends to decrease the predictive skill of the deep learning system, except for 2 meter temperature for 2-day lead time and the past trend for 3-day lead time. Removing the current ice chart has the highest impact on performance (mean +7.14 km on average for all lead times), reducing the skill of the model below that of persistence. However the impact of removing ice



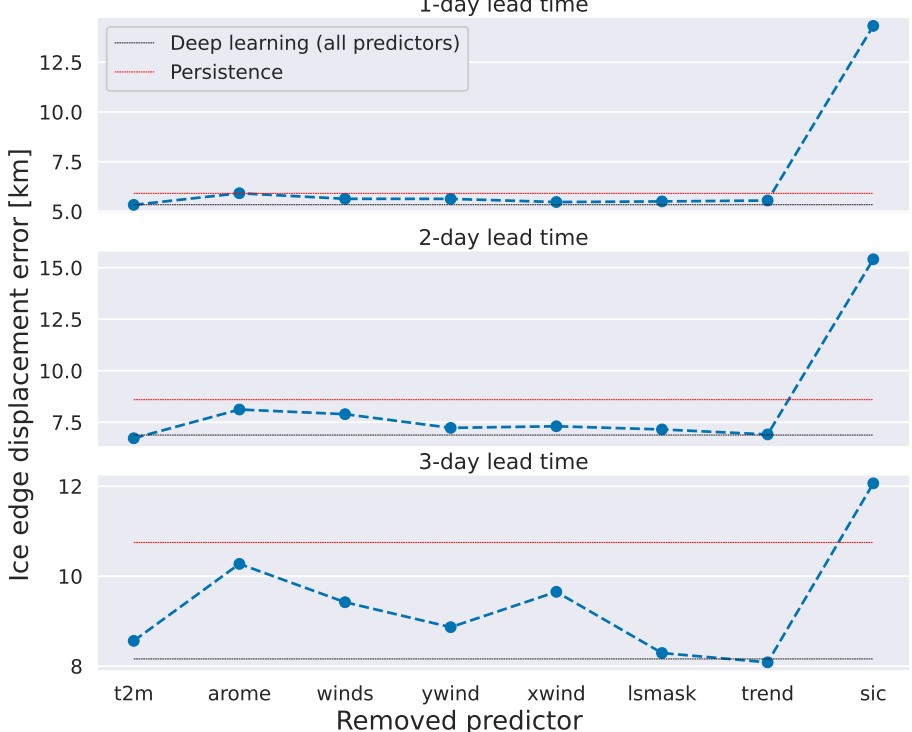

**Figure 10.** Yearly mean nIIEE when a subset of the predictors is withheld during training. The black dashed line denotes yearly mean nIIEE for deep learning forecasts from a model with all predictors, and the red dashed line denotes the skill of persistence. Arome refers to the removal of all atmospheric predictors during training. Winds is similar but for the two wind components.

charts is reduced for increasing lead times. Contrarily, the loss of skill associated with removing all AROME Arctic predictors increases with lead time. Although no other combination of held-back predictors decreases the skill of the deep learning forecasts below persistence, removing all atmospheric forecasts has a consistent negative impact to forecast skill (+1.31 km on average) more than any other removed set, except SIC. Comparing the impact of the different predictors originating from AROME Arctic shows that removing both wind components simultaneously has a greater effect on forecast skill (+0.86 km)
on average than removing 2-meter temperature (+0.08 km). Models trained without the past sea ice trend perform comparably to default deep learning models (+0.06 km).

We also conducted a permutation feature importance analysis to quantify the importance of each predictor for a deep learning model trained on all predictors. Permutation feature importance involves randomly shuffling the input sequence of a single
predictor, and analysing how much this alters the predictive skill of the model. To minimize the potential impact of a seasonal cycle appearing in the reordered predictors, the experiment was run 10 times for each predictor. Permutation feature importance is model specific, and does not provide insight into the predictive capabilities of the analysed predictors. Fig. 11 shows the predictor importance evolution over increasing lead times as the difference in ice edge displacement error from the reference



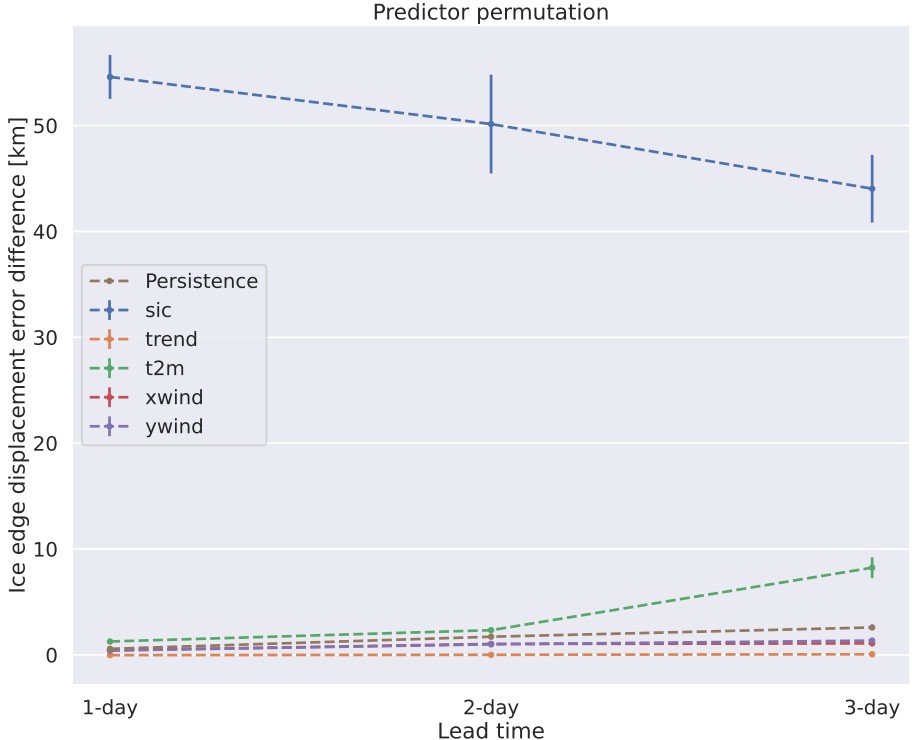

**Figure 11.** Yearly mean nIIEE where the sequence of a predictor in the test dataset has been shuffled, repeated 10 times for all predictors. Each line represents a permuted predictor sequence. Unaltered Persistence forecasts are included as benchmark references. The land-sea mask predictor was excluded from the analysis, as it is static regardless of forecast start-date.

deep learning forecasts. Although the importance of each predictor varies with lead time, the order of importance is consistent
between all lead times with the recent ice chart being the most important predictor, near surface temperature ranking second and finally the two wind components ranking about equal as the third most important predictors. Only permuted ice charts and near surface temperature significantly decrease deep learning forecast score below the benchmark skill of persistence. Only ice charts and 2 meter temperature at 3-day lead time attained a noticeable standard deviation ($\geq 0.1$ km) from inputting predictors from different dates. There is an inverse proportional relationship between the importance of the recent ice chart (decreasing)
and the importance of the atmospheric forecasts (increasing) when targeting longer lead times, indicating that the model is more reliant on the future state of the predicted system (atmospheric forecasts) rather than the initial state (recent ice chart) for longer lead times. Hence Fig. 11 suggests the existence of a limit to the predictive capability gained from providing only current sea ice conditions, similar to how persistence and linear trend forecasts inherently lose skill at longer lead times. The skill difference from past sea ice information encoded in the OSI SAF linear trend is indistinguishable (+0.01 km) from the
performance of un-permuted deep learning forecasts, hence the deep learning forecasts are not dependent on the past state of sea ice regardless of target lead time.



## 5 Discussion and conclusions

This study presents the development of a deep learning forecasting system targeting high resolution (1km) and short lead times
(1 – 3 days) taking into account operational constraints related to the real time availability of data. In order to adequately

resolve the skewed distribution of SIC classes in the ice charts (especially in the MIZ which is crucial for skillful forecasts
ensuring maritime safety (Wagner et al., 2020)), we present a novel reformulation of the decoder from the original U-Net
architecture of Ronneberger et al. (2015) which we name cumulative contours (Eq. 2). The cumulative contours demonstrate
how combining architectural design from Multi-Task learning (Zhang et al., 2014) with task specific additive properties of SIC
intervals positively benefit to deep learning forecasting skill, especially for resolving the intermediate SIC intervals constituting

the MIZ. With this reformulation of the U-Net, the deep learning forecasts are able to consistently outperform both baseline
forecasts as well as operational short-range dynamical sea ice forecasting systems (neXtSIM-F and Barents-2.5) in terms of
achieving the lowest ice edge displacement error when considering the ice charts as reference.

Despite training deep learning models to predict SIC conditions from the ice charts only, the deep learning forecasts behave
similarly to baseline-forecasts when validated against independent AMSR2 SIC observations (Spreen et al., 2008) for concen-

tration contours $\geq 40\%$. The increase in deep learning performance seen between the 10% and 40% concentration contours
may be indicative of a shift in SIC distribution for lower concentration values between the two products, as further indicated
by the decreased displacement difference between AMSR2 and the ice charts when considering the 40% compared to the 10%
concentration contour. It is noted that the ASI sea ice retrieval algorithm exerts larger uncertainties for lower concentrations
(Spreen et al., 2008), whereas SIC $< 10\%$ is visible in SAR and optical satellite images used by the ice analysts. However ice

charts are influenced by human decision-making especially in the medium concentrations (40 – 70%) of the MIZ (Dinessen
et al., 2020), which may be sources for ice edge location discrepancy between the two products. The overall performance re-
gardless of reference product suggests a degree of consistency for the developed forecasts between the two reference products.
However the analysis also suggests that inherent differences between sea ice products is reflected by deep learning forecasts,
as the models are trained only to minimize the statistical error of their target sea ice product.

The results from the forecast intercomparison analysis demonstrates that the deep learning forecasts meet the requirements
for forecast accuracy, while considerably reducing the computing time. However, the results from the analysis could be influ-
enced by the uneven sample sizes used for verification at different lead times. Hence we recommend evaluating the forecasts
with longer time series when they become available. Due to the development of the operational weather prediction system
AROME Arctic, the importance of atmospheric predictors could be reduced without fine-tuning of the deep learning models.

With regards to operationalization, the input data supplied to the deep learning forecasting system have been chosen with
considerations of publishing time, with a special constraint for AROME Arctic being the 66 hour forecast length. The current
setup allows three day forecasts to be published every weekday, sent to maritime operators in advance of their valid date and
cover Saturdays and Sundays when ice charts are not produced.

The predictor importance analysis suggests that the deep learning models benefit from an increased and diversified dataset

by increasing the precision of the predicted sea ice edge by 1.31 km when atmospheric forecasts from AROME Arctic (Müller





et al., 2017) are included as predictors. The inclusion of forecast predictors from weather forecasts has previously been shown to increase predictive skill (Grigoryev et al., 2022; Palerme et al., 2023), which further motivates the inclusion of other forecasted physical forcings affecting the sea ice as predictors. We recommend further work to investigate currently unexplored metocean forcings such as ice-wave interactions (Williams et al., 2013) by including fields such as forecasted wave height and wave direction. However, expanding the dataset towards past temporal regimes by including a linear SIC trend derived from OSI SAF observations was shown to have a marginal effect on the forecast skill, indicating that the deep learning models were unable to infer sea ice growth / decline from past observations (Fig. 11), in line with the results of Palerme et al. (2023).

When all predictors were provided as inputs to the deep learning models, the skill of the forecasts was particularly sensitive to the initialization date of the inputted ice chart (Fig. 11). This suggests that a large part of the inferred physics and seasonality originates from the ice charts, which can also explain why the atmospheric predictors are not essential to outperform persistence. Moreover when considering the initialization time of the AROME Arctic predictors, the lessened impact of the atmospheric predictors could also be associated with AROME Arctic not covering the beginning of the forecast period, especially for shorter lead times. Nevertheless as the model sensitivity to the current ice chart tends to decrease for longer lead times, understanding how the model utilizes the increasingly important forecast predictors should be considered, especially when targeting longer lead times. Other works have investigated the use of explainable artificial intelligence methodologies for interpreting climate-science deep neural networks models and results (e.g. Toms et al., 2020; Ebert-Uphoff and Hilburn, 2020; Bommer et al., 2023). This should be given more attention as they present an opportunity to develop new tools for diagnosing machine learning sea ice forecasting systems.

*Code and data availability.* All code necessary to deploy the developed deep learning models, as well as pretrained weights, are available on the following GitHub repository: https://github.com/AreFrode/Developing_ice_chart_deep_learning_predictions. AROME Arctic (https://thredds.met.no/thredds/catalog/aromearcticarchive/catalog.html) and Barents-2.5 (https://thredds.met.no/thredds/catalog/barents25km_files/catalog.html) forecasts, as well as OSI SAF SSMIS sea ice concentration observations (https://thredds.met.no/thredds/catalog/osisaf/met.no/ice/conc/catalog.html) can be downloaded from the MET Norway thredds Data Server (missing Barents-2.5 data can be provided upon request). The ASI AMSR2 sea ice concentration observations are available on the University of Bremen Sea Ice Remote Sensing data archive (https://data.seaice.uni-bremen.de/amsr2/asi_daygrid_swath/n6250/). Gridded Norwegian Ice Service ice charts and neXtSIM data can be provided upon request.

## Appendix A: Comparing nIIEE for high- and low resolution sea ice concentration

In order to evaluate 1 km resolution sea ice forecasts using the ice edge displacement error as derived by Melsom et al. (2019), we assess the validity of applying the metric for high resolution sea ice forecasts by comparing against a coarse resolution (10 km) reference case. We compute nIIEE from the ice charts at 2-day lead time persistence, with ice charts at 1 km resolution as well as downsampled onto a 10 km grid covering the period 2019 – 2020. Mean monthly nIIEE for both forecasts are shown



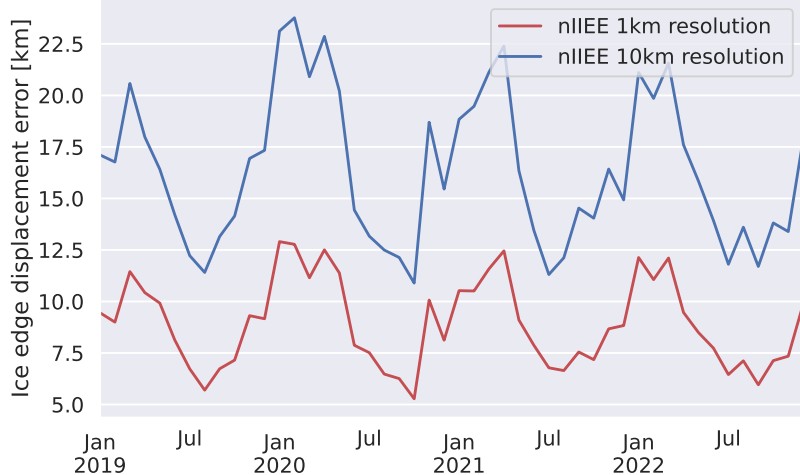

**Figure A1.** nIIEE computed across the entirety of the training dataset (2019 – 2022) for 2-day lead time ice chart persistence with the ice charts as reference. The sea ice edge length used to divide the compute IIEE was derived from the same resolution as the respective forecast.

in Fig. A1. The correlation coefficient between both nIIEE curves in Fig. A1 is 0.98. The strong correlation indicates that the nIIEE is preserved when used in a 1 km resolution environment.

*Author contributions.* A.F.K. conceptualization, analysis, methodology, original draft preparation. C.P. conceptualization, analysis, method-
420 ology, review & editing, supervision. M.M. conceptualization, analysis, review & editing, supervision. J.R. conceptualization, analysis, review & editing. N.H. gridded ice charts, review & editing.

*Competing interests.* The authors declare that they have no conflict of interest

*Acknowledgements.* This work has been supported by the DigitalSeaIce – Multi-scale integration and digitalization of Arctic sea ice observations and prediction models project, which is funded by the Research Council of Norway under contract 328960. C.P. acknowledges support
425 from the SEAFARING project supported by the Norwegian Space Agency and the Copernicus Marine Service COSI project. Copernicus Marine Service is implemented by Mercator Ocean in the framework of a delegation agreement with the European Union. J.R. gratefully acknowledges the support by the Research Council of Norway through the MachineOcean project (grant No. 303411). The authors would like to thank Julien Brajard for constructive discussions.



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
