# Peer review of "Developing a deep learning forecasting system for short-term and high-resolution prediction of sea ice concentration"

_EGUsphere, 2023_

## Author Comment (AC1)

**Response to first referee comments**

We thank the reviewer for their useful comments on our manuscript. Our answers to the comments and modifications to the manuscript are provided below.

**The manuscript addresses the critical need for accurate sea ice forecasting in the Arctic, driven by the increasing maritime activity due to sea ice retreat. A deep learning approach is developed that leverages operational atmospheric forecasts, ice charts, and satellite data to enhance short-term sea ice concentration forecasts within a 1 to 3 days timeframe, aiming for a detailed 1km resolution. The model's performance, validated against various thresholds of sea ice concentration contours, outperforms both baseline forecasts and two state-of-the-art dynamical sea ice forecasting systems across all considered lead times and seasons.**

**Nonetheless, the paper could stand to delve deeper into the model's limitations. Addressing potential biases from the training data and the effects of missing or inaccurate data could enrich the study. Suggestions for improvement are listed as below.**

**1. Place Table 1 within the 'Data' section for better context.**

The table-positioning parameters have been updated to ensure that Table 1 is placed within the 'Data' section.

**2. On page 6, line 140, provide clarification regarding the significance of the 'timeliness of 2.5 hours' for the AROME Arctic model, a detail omitted in Section 2.2.**

We have modified the following sentence in Section 3.1:

*In addition, AROME Arctic has a production time of about 2.5 hours, which ensures that the forecast initiated at 18:00 UTC are available before midnight, allowing us to publish deep learning forecasts on the same day as the input ice chart is published.*

**3. Using operational atmospheric forecasts, ice charts, and Sea Ice Concentration (SIC) from passive microwave observations as predictors is innovative. However, the paper should consolidate potential biases in these data sources and their impact on model performance in the discussion, making the article more logical and complete.**

In the manuscript, when describing AROME Arctic we make sure to address that the system is operational and thus routinely receive updates which impacts distributional properties of predicted variables without retroactive effects. We also further described our choice of limiting training data to 2019 and onward as a direct response to avoid training on channels with differently distributed data. We have modified the following sentence in Section 2.2:

*AROME Arctic has been in operation and continuous development since October 2015, routinely receiving updates which introduces permanent bias changes for predicted variables. Due to a major change to the representation of snow over sea-ice in 2018, a warm bias in near-surface temperatures above sea-ice was significantly reduced in the model (Batrak and Müller, 2019). Thus we start our training dataset at 2019 to avoid supplying our deep learning model with samples containing different temperature biases, especially close to the marginal ice zone (MIZ) where the greatest model response to predictors occurs.*

Although Norwegian ice charts have little documentation regarding uncertainty estimation, we considered the comparison against AMSR2 as an analysis of the sensitivity to the sea ice product used for the target. Figure 5 shows that ice charts and AMSR2 have different occurrence frequency for different thresholds, and we show in our manuscript that initial differences between sea ice products are inherited by our deep learning system. We have modified Section 5 with the following to highlight this result:

*However the analysis also suggests that inherent differences between sea ice products is reflected by deep learning forecasts, and we can not expect the forecasts to improve beyond that initial difference as the models are trained to only minimize the statistical error of their target sea ice product.*

Yet, we disagree that addressing biases will strengthen our analysis. Since deep learning models learn to minimize the output error based on its input, as long as the data is consistently distributed over time, any biases would not impact performance since the model learn those as well. If distributional properties significantly changes in the training data, samples may contribute negatively or be neglected during training overall reducing the skill of the trained network. However as long as the data has a close to constant bias, all samples will contribute positively to the training as the relationship between output and the bias is part of what the model is being taught. We have modified the discussion (Section 5) to address the need to validate deep learning model performance for longer periods of time, since we believe that understanding how updates to physical models supplying predictors to a deep learning system impacts performance is crucial when considering operationalizing machine learning models.

*Hence we recommend evaluating the forecasts with longer time series when they become available. With respect to the development of the operational weather prediction system AROME Arctic, a continued forecast evaluation can also facilitate understanding model*

*response to continuously updated atmospheric predictors and the potential of fine-tuning deep learning models.*

**4. On page 9, line 205, explain the rationale behind the selection of a specific number of epochs for model training.**

We have added the following line to Section 3.3 in the manuscript:

*We chose to train for 25 epochs as the validation loss rarely improved beyond that point.*

**5. The impact of hyperparameter tuning on model performance should be discussed. Were any automated hyperparameter optimization techniques like grid search or Bayesian optimization used?**

We have modified section 4.1 Training performance and data considerations with a specification of what hyperparameters our grid search analysis was performed across.

*The optimal U–Net width of 256 channels in the bottleneck was determined by performing a grid search on the validation dataset across learning–rate (0.0001 - 0.01) and U-Net depth (256 - 1024) (see Figure S2 in the supplement). To achieve consistent architectures between the developed models, we considered only variations of the 2-day target lead time model for the grid search and reused the results for models targeting all lead times.*

We have also added the results from the grid search (Fig. R1) to the Supplement.

**6. In section 4.2, the comparison with dynamical models should include a discussion on the computational efficiency of the deep learning model. This is particularly important for operational forecasting, where timely predictions are crucial.**

We agree with the reviewer, and have added the following to section 4.2 addressing production time of Barents-2.5 in comparison with the deep learning forecasts:

*Comparatively, a single member of Barents completes a 24-hour forecast in $\approx$1min, resulting in a 90% speed up when running on comparable hardware.*

**7. It would be beneficial to conduct a more detailed analysis of the model's performance across various sea ice concentration ranges in Section 4.2.**

We agree with this comment, and have modified Figure 6 to present the Mean annual forecast error across different concentration thresholds (10, 40, 70 and 90%), similar to Figure 8 and Figure 9. The description of Figure 6 in Section 4.2 of the manuscript has also been updated to reflect this change:

*We initially compare the deep learning forecasts against the baseline and dynamical forecasts in 2022 across all target lead times where we consider the yearly mean of the nIIEE for*

[Figure]

Figure R1: Grid search across varying learning rates and bottleneck widths for a deep learning model targeting 2-day lead time. The scores represent the minimum validational loss achieved before terminating training at 25 epochs.

*different sea ice edge contours defined by (10, 40, 70 and 90%) concentration thresholds in Fig R2. For all considered lead times and concentration thresholds, the deep learning forecasts achieves the lowest nIIEE. Similar to persistence, nIIEE for the deep learning forecasts increases proportionally with lead time, although at a lower rate. Additionally, neither neXtSIM, free-drift nor the linear trend forecast are able to outperform persistence, on average for the 10% concentration contour scoring a factor of 1.57, 1.12, and 1.34 higher than persistence, respectively. Furthermore, the mean nIIEE between forecasts based on ice charts (Deep learning, Persistence and free-drift) and NeXtSIM and the linear trend whom are forced by a different sea ice concentration source is notably shifted from the 70% concentration thresholds and above. The nIIEE does not increase much with lead time especially for NeXtSIM when considering higher concentrations.*

[Figure]

Figure R2: Mean annual ice edge displacement error as function of lead time for different sea ice concentration contours defined by 10, 40, 70 and 90% SIC. Only products with a complete coverage of 2022 have been considered.

*The deep learning forecasts improve upon persistence by reducing the $nIIEE_{10\%}$ by a factor of 0.82. In terms of error-growth as a function of lead-time, the linear trend forecast is the only forecast where the slope of the error increases with increasing lead-time regardless of concentration threshold. This indicates that the linear trend from past OSI SAF SSMIS observations is unable to capture ice chart evolution especially for longer lead times. Moreover, the neXtSIM forecasts have the lowest error-growth with lead-time for lower concentrations, indicating that neXtSIM may provide more useful MIZ forecasts at longer lead-times.*

**8. Certain figures, especially those illustrating the model's performance compared to baseline and dynamical models, could be enhanced for clarity and aesthetics. For example, Figure 9 may require modifications to improve clarity.**

We agree that Figure 9 is difficult to interpret and requires modifications to enhance its clarity. We have remade Figure 9 following the styles of Figure 6 and Figure 8, which preserves the content of the figure.

[Figure]

Figure R3: Model intercomparison for varying seasons, lead times and concentration contours. The ice charts are considered as reference. The values reported represent the integrated ice edge error normalized according to the length of the current SIC contour from the reference ice chart in km. The OSI SAF linear trend is computed from the past five days. Barents-2.5 results are only shown for summer and fall.

---

## Author Comment (AC2)

**Response to second referee comments**

We would like to thank the reviewer for their useful comments on our manuscript. In our response we answer the comments made and highlight changes in our manuscript.

**The authors propose a novel operational-like short-term sea-ice forecasting system based on deep learning. Based on past sea-ice charts, satellite images, and weather forecast data, neural networks are trained to predict sea-ice charts for one to three days in advance. To train the neural networks and tackle the issues of unbalances between the sea-ice concentration categories, the authors introduce a new formulation for the categorical prediction. They show that their proposed deep learning system can outperform baseline methods as well as prediction systems based on geophysical sea-ice models.**

**Generally, the approach is sounding and the manuscript follows a logical order. However, the readability of the manuscript can be improved, please see also my minor comments. Additionally, I have a few general comments that should be addressed, before I can recommend an acceptation of the manuscript::**

**1. The proposed method performs better than the second-best method, a Eulerian persistence forecast. Additionally, the two different employed feature importance metrics indicate that the initial sea-ice chart is the most important predictor. From my experience, the shown difference between the deep learning method and persistence could be explained by advection of the sea ice. So, one could wonder how an advection-based (Lagrangian persistence) model would perform in these settings. Based on the wind velocities given by the AROME forecasts, the free-drift equations can be applied to obtain sea-ice velocities, then useable to advect the sea-ice concentration. My feeling says that this might work similarly well as the deep learning method.**

**Even if such a free-drift model would perform similarly to deep learning, this wouldn't mean a shortcoming of deep learning: it would suggest that deep learning can learn such advective behavior without ever seeing any physical relationship. Additionally, deep learning has the potential to exceed this performance with further technological advancements, while the potential for improvements in a free-drift model might be very incremental.**

**An implementation of this might be outside the scope of the article. Nevertheless, I would like to see a discussion of this point in the manuscript and a**

**further reasoning why deep learning outperforms persistence.**

We appreciate the suggestion and agree with this comment that it is of interest to compare our deep learning approach to an advection-based forecast where the sea ice motion is calculated as a response to the atmospheric forcing. As a result, we have included a third baseline forecast based on wind-driven free-drift to our model intercomparison analysis. The wind-driven sea ice motion estimations follow the approach described in Zhang et al. (2024), and uses the same AROME Arctic wind fields as supplied to the neural network. Consequently, Figure 6, Figure 8 and Figure 9 have been updated with the free-drift baseline, where Figure 6 and Figure 9 includes free-drift of the ice charts whereas Figure 8 includes free-drift of AMSR2.

We have modified section 3.5 Baseline-forecasts to also include a description of the free-drift baseline-forecast

*We compare the deep learning forecasts against three baseline-forecasts, persistence of the observations, linear trend of sea-ice concentration from OSI SAF SSMIS and a purely wind-derived sea ice motion estimation based on free-drift. The baseline-forecasts serve as a lower threshold which the deep learning system must outperform in terms of nIIEE in order to be considered skillful. A persistence forecast involves keeping the initial state of the system constant in time. The baseline-forecast based on the linear trend is created by computing a pixelwise linear trend from the previous 5 days which is used to advance the system forward in time. For clarity, the computed values are bounded to match the valid value range [0, 100]. The use of a linear SIC trend as a baseline forecast has previously been assessed in Grigoryev et al. (2022), where the authors reported that the linear trend consistently achieved a higher Mean Absolute Error than persistence.*

*The wind-driven free-drift baseline-forecast is implemented following the description in Zhang et al. (2024). Hence sea ice motion is estimated to be 2% of the surface wind speed 20% to the right (clockwise) of the surface wind direction. Since the free-drift forecast individually advects sea ice parcels based on limited area wind-forcing, the free-drift forecast is not guaranteed to be spatially consistent as some grid cells might not be covered by sea ice after advection while they are clearly in the sea ice pack. Thus we perform nearest neighbor interpolation after advecting the ice to ensure that the free-drift forecasts are spatially consistent. In order to be consistent with the deep learning models, input SIC is advected with the same AROME Arctic mean surface wind fields also supplied as predictors to the deep learning model.*

We have also detailed the performance of the free-drift baseline in section 4.2 Forecast performance and model intercomparison

*Additionally, neither neXtSIM, free-drift nor the linear trend forecast are able to outperform persistence, on average for the 10% concentration contour scoring a factor of 1.57, 1.12,*

*and 1.34 higher than persistence, respectively.*

*For the contours higher than 10% SIC, Fig.8 shows that AMSR2 persistence, AMSR2 free-drift and the deep learning forecasts on average gradually improve against both neXtSIM and the linear trend, with the deep learning forecast increasing its improvement against neXtSIM for higher contours. The difference between AMSR2 free-drift and AMSR2 persistence can also be seen decreasing for increasing concentration contours, yet AMSR2 free-drift achieves a higher nIIEE than the AMSR2 linear trend considering the 10% and 40% concentration contours.*

And finally in section 5 Discussion and conclusions

*Additionally the comparison made against free-drift SIC forecasts suggests that the deep learning model have learned a relationship between the input predictors and target ice chart which is beyond a sea ice motion estimation linearly proportional to the near surface winds. Although it is unknown how the deep learning model respond to individual predictors, the comparison suggests that the models' ability to learn non-linear relationships in the input data aid in predicting SIC. Moreover, the comparison suggests that inferring thermodynamical properties that allow the model to grow and melt sea ice aids when predicting short term SIC beyond that of advection.*

**2. The comparison to numerical systems is nice and shows the potential of deep learning compared to those systems based on geophysical equations. However, the comparison seems not entirely fair:**

**deep learning starts from perfect initial conditions, while the forecasting systems start from an analysis. In Fig. 6, it can be seen that the neXtSIM-F has a very large initialization error and suffers very much from double penalty effects. Hence, I would like to see an experiment, where the deep learning system is initialized with the sea-ice concentration as seen in neXtSIM-F. This way, both forecasts would have the same initialization error, leading to a fairer comparison. In addition, the comparison to the "perfect" initial conditions case could reveal interesting discussion points, e.g., on the stability of the deep learning system or the impact of worse initial conditions in the neXtSIM system, possibly signifying the importance of an improved analysis product.**

Thank you for this comment regarding comparison against neXtSIM-F. We have further specified details about forcings used in neXtSIM-F, as requested in a following comment, which we hope aid to provide a clearer context to the behaviour of neXtSIM-F. However, we do not agree that initializing our deep learning model trained on ice charts with neXtSIM-F sea ice leads to a fairer comparison as we can not expect the deep learning model to make meaningful predictions under such a framework. We also consider a further

analysis of neXtSIM-F performance to be beyond the scope of our paper.

We would also argue that our comparison is still fair in the sense that we compare neXtSIM-F forecasts with deep learning predictions in a operational-like framework. Hence neXtSIM-F performance is not in our control, and from an operational use perspective we use the best neXtSIM-F data available.

We still found that the reviewer raised an interesting point nonetheless, and have further investigated our deep learning model performance with a different sea ice concentration initialization. We propose an alternative experiment which we consider satisfactory in terms of analyzing the effect of initializing the deep learning system with "unoptimal" initial conditions which are in close agreement with the initial conditions of neXtSIM-F. Given that neXtSIM-F is forced by passive microwave observations, we have created a alternative dataset where the input ice charts are replaced by AMSR2 passive microwave observations, yet the ice chart target are kept intact. We have subsequently trained deep learning models on this new dataset, and measured how it performs in terms of predicting the ice charts. The result of this analysis can be seen in the following Figure, which we have also included in the supplement

Fig. R1 will be included in the Supplement. We have also added the following to Section 4.2 in the manuscript:

*However we also trained deep learning models on AMSR2 passive microwave observations with ice charts as target, and deep learning predictions retained sufficient skill comparable to ice chart persistence yet achieving somewhat higher nIIEE (see the Supplement).*

**3. The writing in the methods part is from time to time ambiguous and the reader can easily lose the thread:**

We appreciate the comment on manuscript readability, and we have addressed all suggestions individually

**Dataset pre-processing and selection:**

**Dataset pre-processing and selection: General description of predictors and times needs several times reading and is still partially unclear.**

We hope that the general readability of section 3.1 Dataset preprocessing and selection have been improved following our manuscript modification based on the comments below.

**l. 133ff: "Lead time ... should not exceed the publication time of the target ice chart (15:00 UTC)." Contradiction to the use of AROME initialized at 18:00 UTC?**

We would like to address our choice of using AROME Arctic initialized at 18:00 UTC, six hours later than ice chart valid time which we in the revised manuscript have specified

[Figure]

Figure R1: Same as Figure 6 in the manuscript, but using deep learning models trained with AMSR2 passive microwave observations as input (ice chart cumulative contours are still used as target). Mean annual ice edge displacement error as function of lead time for different sea ice concentration contours defined by 10, 40, 70 and 90% SIC. Only products with a complete coverage of 2022 has been considered. Ice charts are used as reference product.

to consider as 12:00 UTC. We performed an additional experiment, where we appended six hours of AROME Arctic forecasts starting at 12:00 (ice chart valid time) to AROME Arctic runs starting at 18:00 UTC (these are separate model runs, the data is simply appended). Further processing and training is then performed similarly to the manuscript. With the appended AROME Arctic data, we have covered the missing atmospheric development occurring between the ice charts are valid and AROME Arctic at 18:00 UTC is initialized. The result of this analysis can be seen in Fig. R2 which we have included in the supplement.

We have also added the following line to Section 3.1:

*Moreover, the impact of appending six hours of AROME Arctic initialized at 12:00 UTC to the training data have been tested and shown to have an insignificant impact to model performance (see the Supplement).*

We have also rewritten the following for clarity in Section 3.1:

*We want AROME Arctic forecasts to provide the future state of the atmosphere to the deep learning system, which we set to lead times beyond deep learning initialization time. Hence it follows that the atmospheric forecast should cover the time between input and target ice chart valid time.*

**l. 143ff: Why is the temporal development of the atmosphere between 15:00 to 18:00 UTC missed if AROME is initialized for 18:00 UTC?**

See the above comment regarding appending six hours of AROME Arctic (12:00 UTC) data. We have also rewritten this sentence in Section 3.1 to improve clarity and intentions

*When selecting atmospheric forecasts initiated at 18:00 UTC, six hours of future atmospheric development occurring after ice chart valid time (12:00 UTC) is not included in the atmospheric predictors.*

**Fig. 2: Why not imitating how the sea-ice chart is produced by averaging 00:00 UTC to 15:00 UTC with the numerical systems?**

We agree with this comment, and have redone our model intercomparison analysis with physical systems averaged between 00:00 UTC and 12:00 UTC.

We have modified Section 3.5 Model intercomparison setup to:

*When comparing the deep learning forecasts against both physical models, we use the physical forecasts initiated at 00:00 UTC the day following deep learning initialization. Furthermore, physical models are averaged between 00:00 UTC and 12:00 UTC on the target date of the deep learning forecast. This setup is assumed to moderate spatial variability induced by the lack of a temporal mean.*

**l. 150f: I get the argument that it needs less memory during the prediction, but to estimate mean fields, the data has nevertheless to be loaded.**

Rewritten the following in Section 3.1 for clarity:

*Instead of loading multiple high-resolution AROME Arctic fields during training, we preprocess atmospheric variables during dataset creation to reduce the amount of memory needed to load predictors during training.*

**l. 154f: It doesn't matter if the NN takes temporal structures or not into account. You could provide different timesteps as independent channels to the NN and the NN could extract the needed quantities itself. A stronger argument would be that you do feature engineering by using already aggregated statistics.**

We have modified the sentence in Section 3.1 as follows:

*As well as reducing the memory footprint of each predictor, reducing the time steps into a mean-value field also accumulates the temporal changes of each atmospheric variable into a single predictor. Aggregating statistics at an increasing temporal range causes atmospheric predictors to be dependent on target lead time. Hence deep learning models are trained independently for each target lead time.*

**Land-covered grid points (Page 6, l. 125): Unclear if used within the loss function? Do you use zero padding for U-Net? If yes, why not for the land grid points?**

We have removed the use of "input" before data to better specify that the section describes data preprocessing, and not input data to the neural network. We also specify that nearest neighbor land-masking is performed for both input and target ice charts. See the modified paragraph in Section 3.1 below:

*We perform preliminary computations in order to ensure that the data from different sources are on a common grid. The data preprocessing is performed in two stages. Firstly data not matching the AROME Arctic projection are reprojected. Secondly, for data available at a coarser resolution, nearest neighbor interpolation is performed in order to resample the data onto a 1 km grid. The U-Net architecture requires all predictors to have valid values in all grid cells, however both the input and target ice charts and SIC trend do not consistently represent SIC for land covered grid cells due to their intended unavailability. In order to avoid sharp gradients between sea-ice covered seas and land covered areas in the ice charts and SIC trend, we apply a nearest neighbor interpolation of the local sea-ice conditions to fill in the missing sea-ice concentration over land grid points following Wang et al. (2017).*

**It seems like the treatment of the sea-ice concentration by the use of cumulative contours is novel. It deserves its own subsection, which would improve the readability. Nevertheless, remains a bit ambiguous:**

We agree with the comment made by the reviewer, and have added a new subsection which specifically details the implementation on cumulative contours. The parts in the manuscript detailing cumulative contours have been rewritten into a new subsection (Section 3.2 Cumulative contours):

*Motivated by the skewed SIC distribution between the categories which constitutes the MIZ, we reformulate the target SIC such that each category is defined cumulatively and predicted independently using the six SIC thresholds 0, 10, 40, 70, 90% and fast ice (as shown in Fig.1). Cumulative contours are a novel reformulation of the SIC prediction task which aims to preserve the ice chart category distribution. Our proposed target reformulation redefines a categorical ice chart into separate binary fields each containing SIC equal to or greater than a given SIC threshold. With cumulative contours, we provide our deep*

*learning model binary targets which resolve each SIC category with a greater spatial balance than the multi-class ice chart.*

*The cumulative contours are defined as follows. We define $N$ thresholds $k_n \in [0,1]$ which are ordered from lowest to highest with $N$ being the number of contours we want to predict. Each threshold $k_n$ represents a SIC value and is used to classify an ice chart $S$ into a binary field $C^n$, which we denote a cumulative contour. Each element in $C^n$ is defined with the following equation, where i,j denotes spatial indexes*

$$c_{i,j}^n = \begin{cases} 1 & \text{if } s_{i,j} \geq k_n \\ 0 & \text{if } s_{i,j} < k_n \end{cases} \tag{1}$$

*The target reformulation into cumulative contours reduces the classification task into multiple independent binary predictions. Each cumulative contour includes SIC above a set threshold, ensuring that categories in the MIZ are not underestimated due to underrepresentation in the target dataset. We assume each cumulative contour to be ordered such that $C^{n+1} \subset C^n$, however the deep learning model predicts each cumulative contour independently and can deviate from this assumption. We ensure that the predicted cumulative contours at each grid cell achieve the desired ordering by setting all cumulative contours proceeding a not predicted contour to 0 regardless of the probability assigned by the deep learning model.*

*Finally, the forecasted SIC field $\hat{S}$ is defined as the element wise sum over all remaining predicted cumulative contours:*

$$\hat{S}_{i,j} = \sum_{\text{for all } n} \hat{c}_{i,j}^n. \tag{2}$$

*where each element $\hat{S}_{i,j} \in [0, \ldots, N]$ is a categorical representation of ice chart SIC in increasing order. For this work, we have defined six thresholds $k$ following the six WMO ice concentration intervals used in the ice charts. Thus $\hat{S}_{i,j} = 0$ is "ice free open water" and $\hat{S}_{i,j} = 6$ is "fast ice".*

**How do you estimate the forecasted sea-ice concentration? L. 185 presents the forecasted SIC as sum over all contours. If the contours are between 0 and 1, the sum can be over 1. Do you mean instead the mean? If yes, how do you ensure that the neural network is continuous, meaning what do you do if threshold 50% is predicted and 30% is predicted but 40% has a very low probability? Can happen because of independent predictions.**

The point raised by the reviewer about continuity of the output is good, and we have taken care to describe it in the manuscript in the cumulative contours subsection. See the answer to the comment posted above.

**Important citation about different loss functions for the sea-ice concentration is missing (Kucik and Stokholm, 2022). How does the present study fit into their results?**

We appreciate the suggestion to expand our analysis in light of the achievements in Kucik and Stokholm (2023). We have added the following sentences to Section 3.2:

*Norwegian ice charts represent SIC in unevenly sized concentration categories, hence we treat the prediction of an ice chart as a classification task. For automated ice charting, Kucik and Stokholm (2023) have reported that the Categorical Cross-Entropy loss function achieves the highest rate of true positive predictions. However, ice charts are heavily imbalanced fields mostly populated with ice free open water (0%) and very close drift ice ($\geq 90\%$), and neural networks trained with Categorical Cross-Entropy tend to prioritize predicting the most frequently occurring classes while making fewer true positive predictions for intermediate SIC categories (Kucik and Stokholm, 2023).*

**Model implementation:**

**Although many details are given, some details remain unknown, e.g., how have you tested different architectures? Have you used the validation dataset for that? For which lead time? If multiple, what happens if you had different results for different lead times?**

We have added the following to section 4.1 Training performance and data considerations to specify how we achieved the current architecture.

*The optimal U–Net width of 256 channels in the bottleneck was determined by performing a grid search on the validation dataset across learning–rate (0.0001 - 0.01) and U-Net bottleneck width (256 - 1024) (see Figure S2 in the supplement). To achieve consistent architectures between the developed models, we considered only variations of the 2-day target lead time model for the grid search and reused the results for models targeting all lead times.*

We have not tested different architectures at different lead times, since we wanted to have comparable and consistent models across multiple lead times which we believe is achieved by sharing hyperparameters between all models regardless of lead time. We assume that performing grid search on the 2-day lead time model provides the most balanced hyperparameters in terms of predicting also 1-day and 3-day lead time. Another way we could have approached hyperparameter selection was to perform grid search on the model with the longest lead-time, with an assumption that the hyperparameters optimizing the model performing the "most difficult" prediction task also are suitable for short lead time models.

**l. 191f: How do you go from 64 to 256: by 64→128→256?**

We have taken care to specify that we double the amount of feature maps at each stage. The manuscript has been modified as follows:

*The encoder is initiated with 64 feature maps, and at each stage we double the number of feature maps.*

**l. 192: The network has at its bottleneck a width of 256 feature maps. The depth of the U-Net is 2, because of three stages. The depth of the whole network is the number of layers.**

We thank the reviewer for the clarifications, and have edited the manuscript to ensure that the correct terminology is used.

**l. 198: The explanation of the shared network can be improved. Have I understood it correctly that the network extracts common features and then the last (output) layer combines the features to the prediction of the contours?**

Added a clarification, based on modified cumulative contours subsection

*Each cumulative contour is predicted independently from a shared signal, and a forecasted ice chart is constructed from Eq.2.*

**l. 200: Should be corrected to "The loss function is computed individually for all contours". For "all layers" can be misleading as it could also mean that the loss function is estimated for each layer within the NN.**

We agree, and have modified the sentence for clarity

*The pixelwise binary cross-entropy loss function is computed individually for all output layer contours, and the resulting loss of the model is the sum over the individually computed losses.*

**l. 203: How much memory has the A100 GPU? There are two version with 40GB or with 80GB.**

We have specified that the A100 GPU version used has 80GB.

**l. 206: Learning rate and weight decay are two separate things. The learning rate determines how much of the gradient is added to the weights, and weight decay specifies the amount of normalization each weight experiences. Consequently, the learning rate cannot be weight decayed but just decayed.**

We appreciate the clarification, and have removed the mention of "decay" as it was improperly used in our manuscript. We have modified the manuscript to state that we divide the learning rate by 2 every 10 epoch.

*During training, we use the ADAM optimizer (Kingma and Ba, 2014) with an initial learning rate of 0.001 which we reduce by a factor of 2 every 10 epochs.*

**Since you mention weight decay, was any regularization used for the training of the neural network? If yes, please specify.**

No regularization techniques were explored in this work.

**The section could profit from splitting into several paragraphs, e.g., splitting at line 198 and 202.**

We agree with this comment, and have split the subsection at the specified lines.

**Figure 3: I guess that the target ice chart is only used during training to estimate the loss function and not as input for the U-Net. The figure can be misleading such that the predictors and the target ice chart appear as input into the U-Net? Does the U-Net refine the target ice chart?**

We have remade Figure 3 to highlight that the target ice chart is compared against a predicted ice chart, not the U-Net. We agree that the previous iteration of Figure 3 could cause confusion regarding how the target ice chart was treated by the U-Net. See Fig. R3.

**Minor comments:**

**Abstract: A bit unclear what the target of this study this? Sea-ice charts or continuous sea-ice concentration? The abstract could profit from slight restructuring, e.g., what makes this study important?**

We have specified that the target of this study is to predict sea ice charts at 1 km resolution and $1 - 3$ day lead time.

**Abstract, l. 5: with (future) ice charts as ground truth.**

We agree and have implemented this comment.

**Introduction: Many citations about past deep-learning-based approaches are given, however, be careful about what was forecasted. Some of the studies predict sea-ice concentration categories, others the sea-ice extent, while some also predict the sea-ice concentration directly. The introduction could profit from focusing on the mist important approaches, e.g., how much are they comparable to the here presented study? My suggestion would be to make it either clearer what has been forecasted in those studies or to concentrate only on studies that have the highest similarity to the task of predicting sea-ice charts.**

We want our introduction to cover a wide range of the different deep learning sea ice

forecasting approaches available in the literature, to highlight that there are multiple viable approaches to predicting Arctic SIC. We do not see major discrepancies between regressional and classification approaches when it comes to model behaviour, and such would not like to limit the introduction to only classification networks. Rather, we wanted to frame the introduction by what type of data is used to predict SIC, starting from approaches utilizing reanalysis data targeting seasonal to monthly timescales and then focussing on networks with a short-term objective utilizing forecasting-data. The latter being our approach.

We agree with the comment to clarify what different models have predicted, and have updated the introduction thereafter.

**l. 73: what means high spatial resolution?**

We have specified that we mean ($< 1$km) as high spatial resolution.

**l. 75: how are the ice charts gridded? also how are other products interpolated to the target resolution?**

We have modified the sentence detailing gridded ice charts in Section 2.1:

*For this study, the ice charts have been gridded from vector polygons onto the model domain with a 1km spatial resolution using nearest neighbor.*

Preprocessing of the other products are detailed in Section 3.1 Dataset preprocessing and selection

*We perform preliminary computations in order to ensure that the data from different sources are on a common grid. The data preprocessing is performed in two stages. Firstly data not matching the AROME Arctic projection are reprojected. Secondly, for data available at a coarser resolution, nearest neighbor interpolation is performed in order to resample the data onto a 1 km grid.*

**l. 100: AROME covers most of the ice chart domain? What happens for grid points without AROME cover?**

We have modified this sentence in Section 2.2 and removed the used of "most" since AROME Arctic covers the entire model domain (which is constructed as the intersection between AROME Arctic and the ice charts) and thus all grid points are covered by AROME Arctic.

*We use AROME Arctic forecasts as predictors for this study due to its high spatial resolution and regional coverage of the European Arctic*

**Page 5: Has neXtSIM-F been running with Arome as atmospheric forcing? If not, what is the forcing for neXtSIM-F? Information missing.**

We have added missing information on neXtSIM-F forcing

*NeXtSIM receives oceanic forcing from TOPAZ4 (Sakov et.al., 2012) and atmospheric forcing from ECMWF IFS (Owens and Hewson, 2018)*

**l. 158f: Years are wrong compared to the data period (2019-2022) and Table 2, wouldn't it be rather: "We further split the data such that 2019 and 2020 is used for training, 2021 for validation, and 2022 as test dataset."?**

We thank the reviewer for noticing the mistake in the text. Table 2 is correct, and the manuscript have been corrected accordingly.

**l. 240f: Why not averaging the physical models like the ice chart is created, by taking 00:00 UTC up to 15:00 UTC into account?**

We agree with this comment, and have edited the manuscript to specify that average physicals models between 00:00 UTC and 12:00 UTC, to better match the period covered by the ice charts. We also re-did the model intercomparison analysis with these changes.

**l. 248: Width and not depth of the neural network.**

This has been corrected.

**l. 250ff: Please specify that this information is not shown in the paper. Why not putting a proper ablation study into the Appendix of the paper?**

We have modified section 4.1 Training performance and data considerations to specify that this can be seen in the appendix:

*We compared model implementations without cumulative contours (single output, multi–class segmentation with categorical cross-entropy loss) against deep learning models reformulated with cumulative contours, and we got a better preservation of intermediate contours with the model predicting cumulative contours, especially at longer lead times (see the Supplement).*

And we have also added the following figure to the supplement

**Figure 7: please use different colors than for Fig. 6, otherwise might be a bit misleading.**

We agree with the reviewer, and have updated Figure 7 to use a distinct colormap.

**Figure 7: Check if really because of smoothing: You could for example show the impact of smoothing in the true data on the ice edge length.**

We appreciate the comment, and we wanted to confirm this statement based on the suggestion made by the reviewer.

To confirm that smoothing causes a smoother ice edge, we measured the impact of applying a mean-value filter successively to an ice chart by calculating the ice edge length at each smoothing iteration. The mean-value filter is of size $7 \times 7$ km. We see in Fig. R5 that the ice chart has a lower ice edge length after applying the mean-value filter twice. This Figure will be included in the supplement.

**References**

Kucik, A. and Stokholm, A.: AI4SeaIce: selecting loss functions for automated SAR sea ice concentration charting, Scientific Reports, 13, https://doi.org/10.1038/s41598-023-32467-x, 2023.

Zhang, L., Shi, Q., Leppäranta, M., Liu, J., and Yang, Q.: Estimating Winter Arctic Sea Ice Motion Based on Random Forest Models, Remote Sensing, 16, 581, https://doi.org/10.3390/rs16030581, 2024.

[Figure]

Figure R2: Same as Figure 6 in the manuscript, but using deep learning model trained with additional AROME Arctic data initialized at 12:00 UTC appended to the dataset (data between 12:00 and 18:00 appended). For the 10% concentration contour, the deep learning model with additionally appended AROME Arctic data achieves $(5.127, 6.839, 8.371)$ $\text{nIIEE}_{10\%}$ for each lead time (For reference, the deep learning system considered in the manuscript achieved $(5.307, 6.820, 8.112)$ at the same contour). Mean annual ice edge displacement error as function of lead time for different sea ice concentration contours defined by 10, 40, 70 and 90% SIC. Only products with a complete coverage of 2022 has been considered. Ice charts are used as reference product.

[Figure]

Figure R3: Overview of the input and output to the deep learning forecasting system. The predictors are constructed from individually preprocessed sources, and provided to the network together with an associated target ice chart.

[Figure]

Figure R4: Same figure as Figure 5 of the manuscript, but using deep learning models with a single output layer (and not cumulative contours). Seasonal distribution of each SIC category for 2022 as fraction of total mean SIC area for AMSR2, ice charts and single output layer deep learning models at 1 –3 day lead time. The single output layer models are fitted with the same input data and hyper parameters as the deep learning models predicting cumulative contours, however instead of predicting cumulative contours the models predict ice charts directly and compute the categorical cross entropy loss function directly.

[Figure]

Figure R5: Impact to ice edge length when an ice chart (03-01-2022) is successively smoothed by a mean-value filter. A reference deep learning ice chart edge for a 1-day lead time model is shown horizontally as reference

---

## Author Response (AR2)

**Response to first referee comments**

We thank the reviewer for their positive and constructive comments on our manuscript. Our answers to the comments are provided below.

**I see the authors made revisions of the original manuscript and addressed my comments. The new version of the manuscript shows significant improvement. However, it would be best for the authors to carefully review the figures once again, as some of them could still benefit from enhanced aesthetics to improve readability.**

We agree with this comment, and have made further efforts to improve the readability and presentation of the figures. Specifically we have moved the legend to the right side of ceratin figures to decrease cluttered elements. Figure 5., 6., 8. and 9. in the manuscript as well as Figure S3., S4 and S5. in the supplement have been updated to reflect this change.

**Response to second referee comments**

We would like to thank the reviewer for their positive comments and useful feedback on our manuscript. In our response we answer the comments made and highlight changes made in our manuscript.

**A deep learning system is introduced to predict regional sea-ice charts. Starting from the last available ice chart, and forced by atmospheric fields from AROME Arctic, it is predicted how the sea-ice concentration contours move within the next three days. Outperforming baseline models and physical forecasting systems for short-term forecasts, the added value of this system is clearly presented.**

**All previous reviewer comments were taken into account, and compared to the first version, even a new baseline (free-drift model) has been implemented. However, the description of this new baseline is lacking some important details. For example, what is the numerical integration scheme to advect the particles? Looking into the code, it seems like the implementation is based on advecting the position with a forward Eulerian integration. The concentration is then interpolated with nearest neighbours from the advected positions. Yet, such an integration scheme can lead to diffusion (cf., Fig. 10 in Germann and Zawadzki, 2002). Hence, my suggestion would be to add two or three sentences detailing the implementation and its potential weaknesses, such that the reader can better understand the baseline model, while leaving the results as they are. Beside this, the manuscript is publishable as it is.**

We wish to thank the reviewer for this insightful comment on the description of the free-drift baseline, and we agree that further detailing the free-drift implementation improves manuscript readability.

We have modified Section 3.5 in the manuscript with a detailing of which integration scheme we have used and how it is applied, as well as addressing the weaknesses as they are described in Germann and Isztar (2002). Additions to the manuscript are highlighted in blue.

*The wind-driven free-drift baseline-forecast is implemented following the description in Zhang et al. (2024). Hence sea ice motion is estimated to be 2% of the surface wind speed 20 degrees to the right (clockwise) of the surface wind direction.* *New positions are calculated by advecting each grid cell with it's corresponding wind-speed using a first order*

*forward Euler integration scheme.* Since the free-drift forecast individually advects sea ice parcels based on limited area wind-forcing, the free-drift forecast is not guaranteed to be spatially consistent as some grid cells might not be covered by sea ice after advection while they are clearly in the sea ice pack. Thus we perform nearest neighbor interpolation after advecting the sea ice to ensure that the free-drift forecasts are spatially consistent. *Additionally, it is described in Germann and Isztar (2002) that simple advection schemes tend to introduce numerical diffusion resulting in a loss of smaller scale features.* Finally, in order to be consistent with the deep learning models, input SIC is advected with the same AROME Arctic mean surface wind fields also supplied as predictors to the deep learning model.

**References**

Germann, U. and Isztar, Z.: Scale-Dependence of the Predictability of Precipitation from Continental Radar Images. Part I: Description of the Methodology, Monthly Weather Review, 130, 2859–2873, https://doi.org/10.1175/1520-0493(2002)130⟨2859:sdotpo⟩2.0.co;2, 2002.